# NEURAL MANIFOLD OPERATORS FOR LEARNING THE EVOLUTION OF PHYSICAL DYNAMICS

## ABSTRACT

Modeling the evolution of physical dynamics is a foundational problem in science and engineering, and it is regarded as the modeling of an operator mapping between infinite-dimensional functional spaces. Operator learning methods, learning the underlying infinite-dimensional operator in a high-dimensional latent space, have shown significant potential in modeling physical dynamics. However, there remains insufficient research on how to approximate an infinite-dimensional operator using a finite-dimensional parameter space. Inappropriate dimensionality representation of the underlying operator leads to convergence difficulties, decreasing generalization capability, and violating the physical consistency. To address the problem, we present Neural Manifold Operator (NMO) to learn the intrinsic dimension representation of underlying operators by calculating the minimum dimensional submanifold representation in the latent space. NMO achieves state-of-the-art performance in statistical and physical metrics and gains 23.35% average improvement on three real-world scenarios and four equation-governed scenarios across a wide range of multi-disciplinary fields. Our paradigm has been demonstrated universal effectiveness across various model structure implementations, including Multi-Layer Perceptron, Convolutional Neural Network, and Transformer. Experimentally, we prove that the intrinsic dimension calculated by our paradigm is the optimal dimensional representation of the underlying operators. Our code is available at https://anonymous.4open.science/r/Neural_Manifold_Operator.

## 1 INTRODUCTION

Modeling the evolution of physical dynamics is the foundation for studying and predicting physical systems, which is a common challenge in science and engineering (Bender, 2000). Throughout the history of science, analytical models of physical dynamics (e.g. Newton's laws of motion) derived from the first principle are used to study the evolution of physical systems and make physical dynamics predictable (Kibble & Berkshire, 2004), which breeds a lot of real-world applications, such as numerical weather prediction systems (Bauer et al., 2015). However, when facing real-world scenarios, such physical systems with high degrees of freedom and complexity make solving the model and quantifying its evolution harder, which generally means higher computational costs and more approximate assumptions to compromise.

With the rapid development of deep learning, a new paradigm for modeling and predicting physical dynamics is widely discussed. Deep learning models can learn underlying physical relationships from data and predict the future state at a lower cost, which leads to many achievements in the study, modeling and prediction of physical dynamics (De Bézenac et al., 2019). Different from other areas in deep learning, learning the evolution of physical dynamics is generally equivalent to learning nonlinear infinite-dimensional operator mappings between Banach space (Temam, 2012), which requires deep learning models to have enough generalization capable of learning the intrinsic dynamics of the physics system, instead of local fitting for training data.

Learning the intrinsic dynamics of physics systems is critical for deep learning models. Although several studies regard physics variables as computer vision tasks and get good performance in statistical metrics, these methods usually get poor performance in physics consistency and are hard to generalize into similar scenarios in the same physics system (e.g. different initial conditions or configure parameters). Recently, operator learning, a class of deep learning methods designed

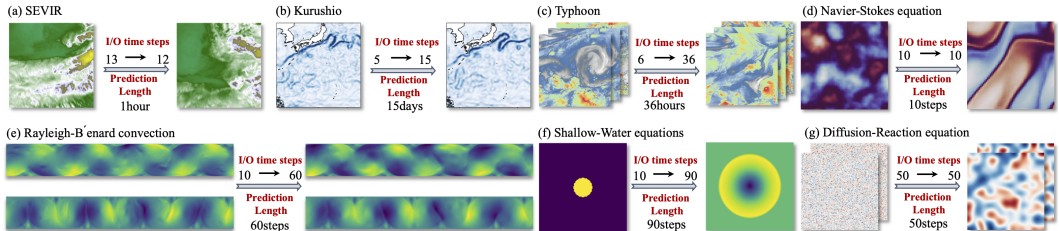

Figure 1: The prediction visualization of NMO in several scenarios.

for learning infinite-dimension operators (Kovachki et al., 2023; Lu et al., 2021), has been widely employed for modeling physical systems. Such methods generally project the original physics space into higher-dimensional latent space and parameterize the underlying operators describing the evolution of physical dynamics by neural network structures. However, the determination of the latent space dimension in these methods is generally subjective and empirical, and even it may change in the same physical system with different configured parameters. However, the redundant dimension representation of the underlying operators leads to several problems including the convergence difficulty, reducing the generalization capability, and even destroying the physics consistency of physical systems. Therefore, how to approximate the infinite-dimension operator using an appropriate finite-dimension parameter space is the key problem for further developing operator learning methods.

To address the problem, we propose Neural Manifold Operator (NMO), an operator learning paradigm for learning the intrinsic dimension representation of the underlying operator. By calculating the minimum dimensional submanifold representation of the variables in the latent space, NMO can adaptively determine the intrinsic dimension of the physical system. By projecting the latent space into a compact space of the intrinsic dimension, NMO enables efficient and accurate learning of the underlying operators and preserves the physics consistency of the system. We introduce several benchmarks, including real-world scenarios and equation-governed scenarios which encompass complex weather and ocean systems, as well as chaotic and interacting physical dynamics, aiming to evaluate the capacity of our model for approximation, generalization, and preserving physical properties. Compared to several baseline models, NMO achieves state-of-the-art performance in statistical and physical metrics. We experimentally demonstrate that the intrinsic dimension calculated by our paradigm is the optimal dimension of the latent space in efficiency and accuracy. Our paradigm applies to various physical systems and different neural network structure implementations.

In summary, our contributions are as follows:

- **Limitations of high-dimensional latent representation:** We analyze several shortcomings of redundant dimensionality of latent space and experimentally demonstrate that the intrinsic dimension is the optimal dimension representation.

- **Generic operator learning paradigm:** NMO is a generic operator learning paradigm for various network structure implementations including Multi-Layer Perceptron, Convolutional Neural Network, and Transformer.

- **Benefits in multi-disciplinary areas:** NMO achieves state-of-the-art performance in several real-world and equation-governed scenarios, ranging from mathematics, physics, chemistry and earth science.

- **Efficiency and Accuracy:** By intrinsic dimension projection, NMO significantly reduces the training parameters and effectively improves the capability of generalization and physical consistency.

## 2 PRELIMINARIES

### 2.1 DEEP LEARNING FOR PHYSICAL DYNAMICS

In recent years, it has been produced a lot of elaborative deep learning methods for learning physical dynamics. Due to the similar tensor shape, modeling physical systems is often viewed as computer vision problems. Several state-of-the-art models designed for computer vision tasks (e.g. image super-resolution or video prediction) are used to model physics dynamics. However, physics inconsistency,

unexplainability, and poor generalization limit further development. More deep-learning methods guided by physics theory have been designed, which can be roughly categorized into equation-constraint, interpretable-structure, and operator learning methods.

**Equation-constraint methods**  Incorporating physical laws into the loss function, physics-informed machine learning methods (Karniadakis et al., 2021) ensure that the prediction result satisfies specific physical properties, and even achieve unsupervised prediction for equation-governed dynamics (Erichson et al., 2019; Wang et al., 2020; Zhu et al., 2021; Shokouhi et al., 2021; Wang et al., 2019; Wang & Perdikaris, 2023). However, for complicated scenarios such as real-world dynamics, incomplete physics laws and imbalance of loss function terms makes optimization for the neural networks significantly hard limit the performance of the models (Rohrhofer et al., 2022; Wang et al., 2022b).

**Interpretable-structure methods**  Interpretable-structure methods for learning physical dynamics use the mathematical equivariance between deep learning structure and physical equations to design architectures, which incorporate more physical inductive bias. PDE-Net (Long et al., 2018) proves the similar mathematical properties of the convolution operation as the difference operator and leverages the theory to develop a framework for learning time-dependent partial differential equations. Neural ODE (Chen et al., 2018) demonstrates that continuous Residual Networks(He et al., 2016) can be mathematically expressed as ordinary differential equations, and is utilized for predicting the dynamics systems (Kiani Shahvandi et al., 2022; Höge et al., 2022; Mehta et al., 2021). Based on Noether's theorem, equivariant deep learning methods incorporate geometric symmetry into neural networks by equivariant group transformation for constrain conservation of physical systems (Gerken et al., 2021; Wang et al., 2020; Dehmamy et al., 2021; Villar et al., 2021; Brandstetter et al., 2022b;a; Walters et al., 2020). However, such models with strong physics inductive bias in structure may not be generic applicable in different physics scenarios, and even degrade the performance and generalization capability of the model in real-world dynamics with noisy or incomplete data (Wang et al., 2022a).

**Operator-learning methods**  Operator learning methods are designed for learning mappings between infinite-dimensional function spaces. Based on the universal approximation theorem (Cybenko, 1989; Hornik et al., 1989), DeepONet (Lu et al., 2021) learns the target operator by sampling the function space. Koopman theory (Koopman, 1931) inspires several methods designed to approximate the infinite-dimension Koopman operator in the observation space (Lusch et al., 2018; Yeung et al., 2019; Xiong et al., 2023a;b). Besides, Green's function-based models convert infinite-dimensional operator mappings into kernel integral parameterization (Li et al., 2020a; Kovachki et al., 2021; Li et al., 2020b; 2021; 2020c; Tripura & Chakraborty, 2022). However, these methods learn the underlying operators in a high-dimensional latent space, but further discussion about the dimension of latent space is still lacking. Accurately representing infinite-dimensional operators in finite-dimensional parameter space remains a challenge.

## 2.2 DIMENSION REPRESENTATION OF OPERATORS

The intrinsic dimension can be conceptualized as the minimum number of variables or parameters required for a minimal representation, which is often regarded as the minimal number of hidden neurons for the deep learning model to represent the target. Estimating the intrinsic dimension of physical systems is good for learning the underlying intrinsic dynamics behind the data (Champion et al., 2019; Floryan & Graham, 2022), finding parameterized surrogate models and building reduced order models (Bai et al., 2005; Lee & Carlberg, 2020; Fresca et al., 2021). For operator-learning methods, finite dimension representation for infinite operators is indispensable. Low-rank Neural Operator (Kovachki et al., 2021) reconstructs r-rank operator by SVD. Dynamic Mode Decomposition (Schmid, 2010; 2022) and several Koopman-based deep learning methods (Yeung et al., 2019; Xiong et al., 2023a;b) have been developed to identify the invariant subspace of the Koopman operator, allowing for finite-dimensional linear representations of complex dynamic systems. NOMAD learns a low-dimensional representation of solution with a nonlinear manifold decoder (Seidman et al., 2022). With a new universal approximation theorem under minimal assumptions for the underlying operator, PCA-Net partially overcomes the general curse of dimensionality for operator learning (Lanthaler, 2023). However, it still lacks a unified paradigm designed for learning the intrinsic dimension representation of operators that is applicable to various physical systems and model structures.

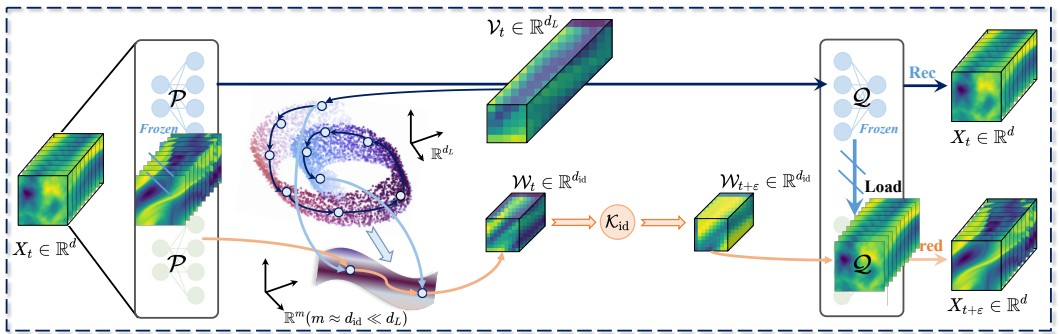

Figure 2: Overview of NMO.

# 3 NEURAL MANIFOLD OPERATOR

## 3.1 PROBLEM DEFINITION

Considered a physics variable $X$ on the bounded $D \subset \mathbb{R}^d$. To study the temporal evolution of the physical variables, it is natural to formulate the dynamical system as

$$\frac{dX}{dt} = f(X). \tag{1}$$

The time evolution of the physics variables $X$ can be represented as

$$X_{t+\varepsilon} = \mathbf{F}(X_t, t) = X_t + \int_t^{t+\varepsilon} f(X_\tau, \tau) d\tau, \tag{2}$$

where the evolution mapping is $f : \mathbb{R}^d \to \mathbb{R}^d$.

The aim of our architecture is that build an approximation of $G$ to parameterize the flow mapping of the physics variables from the finite observation of the physics system by constructing a parametric map

$$G : X_t \times \Theta \to X_{t+\varepsilon} \tag{3}$$

with the finite-dimensional parameter space $\Theta$. Naturally, a series of measurements of physics variables on the bounded $D \subset \mathbb{R}^d$ can be defined as $X_{0:n} = \{X(t, \cdot) : t = t_0, t_1, \cdots, t_n\}$. By defining a loss function and the specific optimization algorithm, the optimal parameter $\theta^\dagger \in \Theta$ can be calculated by a finite collection of observation $X_{0:t}$ and its future state $X_{t:t+\varepsilon}$. Our architecture is designed for learning the infinite operator, which describes the evolution of the physics variable, to generalize the whole physics system in Banach space.

## 3.2 OVERALL ARCHITECTURE

There are three main structures in our architecture: an encoder $\mathcal{P} : \mathbb{R}^d \to \mathbb{R}^{d_L}$ for projecting from physics space to latent space, a decoder $\mathcal{Q} : \mathbb{R}^{d_L} \to \mathbb{R}^d$ for projecting from latent space to physics space, and time evolution operator $\mathcal{K} : \{v_t : D_t \to \mathbb{R}^{d_L}\} \to \{v_{t+\varepsilon} : D_{t+\varepsilon} \to \mathbb{R}^{d_L}\}$ to learn the evolution of physics systems in latent space.

In the pretraining stage, the encoder and decoder are trained by a self-supervision strategy for reconstructing. The goal of the stage is to learn the high-dimensional representation of the physics variable in latent space. When the encoder and decoder are trained to converge, the latent variable $V \in d_L$ is the latent representation of the physics variable. The reconstruction process is achieved as

$$X_t = \mathcal{Q} \circ \mathcal{P}(X_t). \tag{4}$$

After the self-supervision pretraining process for reconstructing, the physics variable $X$ is projected into a latent variable $\mathcal{V}$ in higher dimension latent space $\mathbb{R}^{d_L}$ by the encoder $\mathcal{P}$, which means there exists an m-dimensional Riemann submanifold $\mathcal{M}$ of $R^{d_L}$ to represent the latent variable. The

m-dimensional Riemann submanifold in over-dimensional space can be calculated by the manifold learning algorithm described in Section 3.3.2. The dimension of the submanifold $\mathcal{M}$ can be considered as the intrinsic dimension $d_{\mathrm{id}}$ of the latent variable. Therefore, linear projection $\mathcal{L} : \mathbb{R}^{d_L} \to \mathbb{R}^{d_{\mathrm{id}}}$ are introduced to learn the underlying operator of physics systems in a compact space by the time evolution operator module. In summary, our overall architecture is outlined as

$$G_\theta := \mathcal{Q} \circ \mathcal{L}^{-1} \circ \mathcal{K}_{\mathrm{id}} \circ \mathcal{L} \circ \mathcal{P}, \tag{5}$$

where $\mathcal{K}_{\mathrm{id}}$ is the intrinsic dimension representation of the operator and $\mathcal{L} : \mathbb{R}^{d_L} \to \mathbb{R}^{d_{\mathrm{id}}}$ is the linear projection. By defining a cost function $L$, the optimal parameter to representation for the evolution operators is calculated by solving the optimal question expressed as

$$\min_{\theta \in \Theta} \mathbb{E}\left[L\left(G, G_\theta\right)\right]. \tag{6}$$

## 3.3 Latent Space Projection Network

### 3.3.1 Projection Network Structure

The function of the projection network is projecting between physical space and latent space for the physics variables $X_t \in \mathbb{R}^d$ and latent variables $\mathcal{V}_t \in \mathbb{R}^{d_L}$, which is mainly composed by an encoder $\mathcal{P} : \mathbb{R}^d \to \mathbb{R}^{d_L}$ and a decoder $\mathcal{Q} : \mathbb{R}^{d_L} \to \mathbb{R}^d$. The projection network is trained in the pretraining process by self-supervision latent reconstruction with the L2 loss function:

$$L_{rec} = ||\mathcal{Q} \circ \mathcal{P}(X_t) - X_t||_2^2. \tag{7}$$

The structure of encoder $\mathcal{P}$ and decoder $\mathcal{Q}$ are mainly composed of a series of convolution modules and transposed convolution modules respectively. The detailed structural design can be found in Appendix X.

### 3.3.2 Intrinsic dimension calculation

The latent variables can be viewed as $\mathcal{V} = \{\boldsymbol{V}_1, \ldots, \boldsymbol{V}_n\}$ composed by $n$ independent and identically distributed vectors in latent space $\mathbb{R}^{d_L}$. There exists that the latent variables $\mathcal{V}$ can be constrained into a m-dimensional Riemannian submanifold $\mathcal{M}$ in $\mathbb{R}^{d_L}$, where $m$ is less than $d_L$. The smallest existing $m$ is considered as the intrinsic dimension of the manifold. Therefore, the goal of the manifold algorithm is to calculate minimal $m$.

The latent variables can be seen as the observation points of the submanifold. The methods of calculating intrinsic dimension can be viewed as a question about the estimation of density functional of the observation points (Costa et al., 2005):

$$\log \int_{B(\boldsymbol{v}_0, r)} g(f(\boldsymbol{v}))\mu(\mathrm{d}\boldsymbol{v}), \tag{8}$$

where $g$ is an associated metric of the submanifold $\mathcal{M}$ and $B(\boldsymbol{v}_0, r)$ is the ball with radius $r$ centered at points $v_0$. Given suitable function $g$, the density functional can be approximated by the number of observation points falling into set $B(\boldsymbol{v}_0, r)$. $T_k(\boldsymbol{v}_0)$, the distance from the observation point $v_0$ and its k-nearest neighbor, is related to the choice of $r$ (Pettis et al., 1979). The number of the observation points falling into the k-nearest ball $B(\boldsymbol{v}_0, T_k(\boldsymbol{v}_0))$ can be approximated by a Poisson process, and maximum likelihood estimation can be used to calculate the intrinsic dimension in local as the following form (Levina & Bickel, 2004; Costa et al., 2005):

$$\hat{m_0} = \frac{1}{k-1} \sum_{j=1}^{k-1} \log \frac{T_k(\boldsymbol{v}_0)}{T_j(\boldsymbol{v}_0)}. \tag{9}$$

The intrinsic dimension $m$ of the manifold $\mathcal{M}$ can be approximated by the average of all observation points as (Levina & Bickel, 2004; Chen et al., 2022):

$$\hat{m} = \frac{1}{n} \sum_{i=1}^{n} \hat{m}_k, \quad m = E\left(\hat{m}(\boldsymbol{v})\right). \tag{10}$$

## 3.4 TIME EVOLUTION OPERATOR

Having calculated the intrinsic dimension, the latent variable $\mathcal{V}_t$ is linearly transformed into its compact representation $\mathcal{W}_t$. The time evolution operator $\mathcal{K}_{\text{id}}$ is used to learn the underlying evolution of physics dynamics in compact latent space $\mathbb{R}^{d_{\text{id}}}$. We decompose the time evolution operator into three operators to learn the underlying evolution relationship in various scales. The generic form of the time evolution operator module is

$$\mathcal{K}_{\text{id}} = \mathcal{K}_u \circ \mathcal{K}_e \circ \mathcal{K}_d, \tag{11}$$

where $\circ$ denotes operator decomposition, downsampling operator $\mathcal{K}_d : \mathcal{W}_t \to \mathcal{T}_{\mathcal{W}_t}$, evolution capture operator $\mathcal{K}_e : \mathcal{T}_{\mathcal{W}_t} \to \mathcal{T}_{\mathcal{W}_{t+\varepsilon}}$ and upsampling operator $\mathcal{K}_u : \mathcal{T}_{\mathcal{W}_{t+\varepsilon}} \to \mathcal{W}_{t+\varepsilon}$. According to our theory, an appropriate parameterization space enables neural networks to learn intrinsic dynamics and the theory applies to various structures. Therefore, we design three different implementations for evolution capture operator $\mathcal{K}_e$ based on Multi-Layer Perception, Convolution Networks, and Transformer respectively. When the encoder and decoder network have been trained in the pretraining process and its parameters are frozen, the Time Evolution Operator module is trained by the supervision strategy. The Time Evolution Operator will converge to the underlying evolution operator by minimizing the loss function

$$L_{pred} = ||G_\theta(X_t) - X_{t+\varepsilon}||_2^2. \tag{12}$$

## 4 EXPERIMENTS

**Benchmarks.** As shown in Table 1, we use real-world scenarios and equation-governed scenarios to evaluate our model which includes 7 datasets. Here are the descriptions of these datasets.

(1) **SEVIR** (Veillette et al., 2020) includes satellite and radar weather data, which we use to evaluate our model's accuracy in forecasting short-term severe weather events like thunderstorms and intense precipitation. (2) **Kurushio** is a strong western boundary current, which is a challenge for Earth system modeling and prediction. The Kuroshio stream dataset is the vector data of sea surface stream velocity from the Copernicus Marine Environment Monitoring Service (CMEMS). (3) **Typhoon** (Bessho et al., 2016) is a three-layer water vapor channel dataset covering the East and Southeast Asian Pacific coastal regions. We use it to test the model's ability to predict water vapor distribution in the next 36 hours, thereby achieving preliminary typhoon forecasting. (4) **Navier-Stokes equation** (Li et al., 2020a) describes the dynamics and mass transport of the general fluid. We select the two-dimensional equations for an incompressible viscous fluid with a viscosity coefficient of $10^{-5}$ to test our model for learning complicated fluid dynamics with high Reynolds numbers. The evolution of vorticity is computed from the equation solved by the pseudo-spectral method. (5) **Shallow-Water equations** (Takamoto et al., 2022) describes the fluid in the shallow water approximation and barotropic system, which is often used for large-scale geophysical flows and tsunami simulations. The dataset is well-suited for testing the performance in mass conservation and long-term prediction. (6) **Rayleigh-Bénard convection** (Chirila, 2018; Wang et al., 2020) describes the turbulent flow arises from convection induced by bottom heating, which is the main mechanics of the El Niño and Southern Oscillation. The dataset is simulated by the Lattice Boltzmann Method, which is appropriate for testing the ability of our model to learn turbulence and energy conservation. (7) **Diffusion-Reaction equation** (Takamoto et al., 2022) models the interplay between the diffusion of substances and their chemical reactions, often used to describe processes in materials, biology, and the environment. The dataset, calculated by the standard finite volume solver, is a challenging benchmark due to there are two non-linearly coupled variables, the activator and the inhibitor.

**Baselines.** Several advanced and representative models in computer vision, time series prediction, neural operator and partial differential equations solving, are used for evaluating our model. U-Net (Ronneberger et al., 2015), Residual Networks (ResNet) (He et al., 2016) and Swin-Transformer (Swin) (Liu et al., 2021) are representative and mainstream computer vision backbone model, which is often used for various tasks. SimVP-v2 (Tan et al., 2022), PredRNN-V2 (Wang et al., 2022c) are representative general models for time series prediction. EarthFormer (Gao et al., 2022) is designed for the time series of the Earth system. Fourier Neural Operator (FNO) (Li et al., 2020a) is one of the most representative neural operator models designed for learning mapping between Banach space. Turbulence-Flow Net (TF-Net) (Wang et al., 2020) and Latent Spectral Models (LSM) (Wu et al., 2023) are advanced physics-guided models incorporating physics knowledge into inductive bias.

Table 1: Performance comparison with 9 baseline models in all scenarios. RMSE is used for the evaluation of these models, with a smaller RMSE value indicating greater accuracy. Since FNO is designed for single variables prediction, we only evaluate these models in single variables scenarios to ensure optimal performance of the baseline models. The underline indicates the most accurate result in baseline models. The bold font indicates the most accurate of all models. The asterisk (*) denotes GPU memory overflow (exceeding 40GB). The forward slash (/) indicates that the original model is only designed for single variable prediction.

| MODEL | REAL-WORLD SCENARIOS | | | EQUATION-GOVERNED SCENARIOS | | | |
|---|---|---|---|---|---|---|---|
| | SEVIR | KUROSHIO | TYPHOON | NAVIER STOKES | SHALLOW WATER | RAYLEIGH-BÉNARD CONVECTION | DIFFUSION REACTION |
| U-NET | 2.0280 | 0.0591 | 0.0546 | 0.4451 | 0.0890 | 0.3977 | 0.0612 |
| RESNET | 2.0787 | 0.0709 | 0.1246 | 0.5246 | 0.0730 | 0.5746 | 0.0820 |
| PREDRNN-V2 | 1.9741 | 0.0651 | 0.0234 | 0.5196 | 0.0970 | 2.2965 | 0.1201 |
| SWIN-TRANSFORMER | 2.0067 | 0.1682 | 0.0273 | 0.4741 | 0.0434 | 1.6852 | * |
| SIMVP-V2 | 0.7943 | 0.0658 | 0.0193 | 0.3872 | 0.0098 | 2.3804 | 0.0043 |
| EARTHFORMER | 0.2877 | 0.1612 | 0.0671 | 0.4472 | * | 1.5746 | * |
| TF-NET | 2.1946 | 0.1033 | 0.0172 | 0.4243 | 0.0860 | 0.2076 | 0.0037 |
| FNO | 1.0099 | / | / | 0.2547 | 0.0045 | / | 0.0008 |
| LSM | 1.2569 | / | / | 0.2863 | 0.0087 | / | 0.0009 |
| **NMO** | **0.1698** | **0.0404** | **0.0161** | **0.2487** | **0.0028** | **0.1418** | **0.0007** |
| PROMOTION | 41.01% | 31.64% | 6.40% | 2.35% | 37.78% | 31.74% | 12.5% |

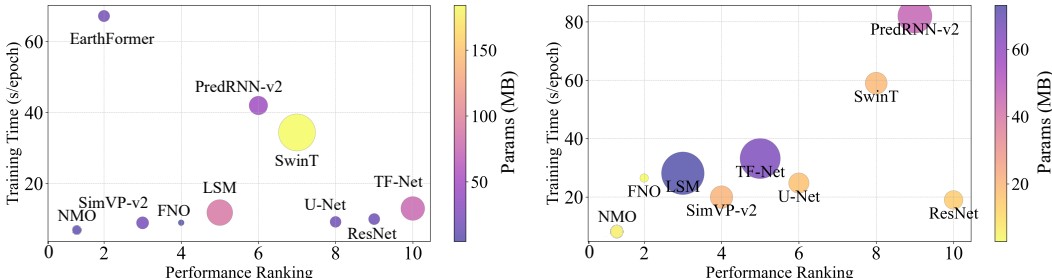

Figure 3: The training time and RMSE performance rankings (from lowest to highest) of various models on SEVIR and Navier-Stokes equation scenario.

## 4.1 EVALUATION METRICS

### 4.1.1 STATISTICAL METRIC

**Root Mean Square Error.** Root Mean Square Error (RMSE) is a widely accepted metric for quantifying the statistical performance of the deep learning model, which can reflect the average error of the prediction result.

### 4.1.2 PHYSICAL METRICS

Although statistical metrics can evaluate the pixel-wise performance of models, more physics metrics are indispensable to evaluate whether models learn the physical properties rather than local fitting. There are three physical metrics in specific scenarios.

**Mass Conservation.** For incompressible shallow water wave equation with free surface and closed boundary, the prediction variable $h$ not only describes the depth of water but also is proportional to the mass of the water column. Therefore, the total mass of the system can be calculated by the variable $h$ to evaluate whether the models preserve first-order conserved quantities. The mass conservation formula of the 2-dimensional shallow water equations can be expressed as

$$\frac{d}{dt} \iint_D h \, dx \, dy = 0, \tag{13}$$

where $D$ is the computational domain of the equations in 2-dimensional Euclidean space $\mathbb{R}^2$.

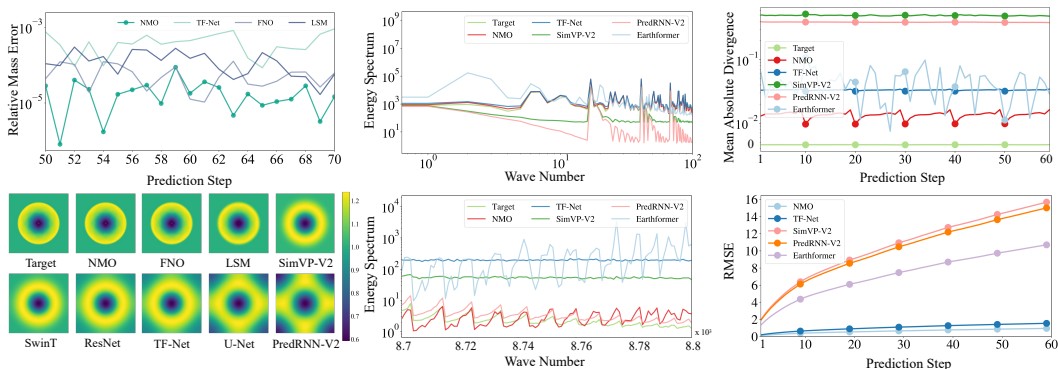

Figure 4: **Left:** Relative mass error at each time step and visualization of prediction results of each model on the Shallow-Water equations scenario. **Mid:** Turbulence energy spectrum on the Rayleigh-Bénard convection scenario. **Right:** The average of absolute divergence convection at each time step and RMSE associated with the prediction step of each model on the Rayleigh-Bénard convection scenario.

**Energy Conservation.** For the Rayleigh-Bénard convection scenario, the turbulence energy spectrum indicates the kinetic energy contained in eddies with wavenumber $k$. The turbulence energy spectrum is calculated by mean turbulence kinetic energy after Fourier transformation. The metric is appropriate for analyzing energy consistency in different ranges of wavenumber. The energy spectral $E(k)$ is calculated by

$$\int_0^\infty E(k)dk = \frac{1}{2T} \sum_{t=0}^{T} [(u_x(t) - \bar{u_x})^2 + (u_y(t) - \bar{u_y})^2], \tag{14}$$

where $u_x$ and $u_y$ is the component of velocity with respect to the x-axis and y-axis, the bar symbol means time average, $t$ is the time step and $T$ denotes the prediction length.

**Divergence.** Derived by the continuity equation, the divergence of velocity $\nabla \cdot \mathbf{u}$ should be zero for the incompressible fluid parcel, which is the closure condition and fundamental constraint of mass conservation in fluid dynamics. Calculating the average of absolute divergence in the whole fluid field as a physical metric at each time step indicates whether the model learns the intrinsic dynamics of fluid transportation. The divergence formula can be expressed as

$$\nabla \cdot \mathbf{u} = \frac{\partial u_x}{\partial x} + \frac{\partial u_y}{\partial y} = 0, \tag{15}$$

where $\mathbf{u} = \{u_x, u_y\}$ is a 2-dimensional velocity vector.

## 4.2 ACCURACY AND EFFICIENCY

As shown in Table 1, we evaluate NMO and 9 baseline models in real-world scenarios and equation-governed scenarios. NMO achieves state-of-the-art performance in all scenarios. On average, NMO outperforms the best baseline method on each benchmark by 23.35%. For real-world earth system scenarios, NMO outperforms the EarthFormer by 63.98% averagely. For fluid dynamics scenarios, NMO outperforms TF-Net with a strong fluid inductive bias by 62.73% averagely. According to Figure 3, NMO not only achieves the best accuracy but also the fastest training speed through a lightweight implementation.

## 4.3 PHYSICAL CONSISTENCY ANALYSIS

As Figure 4 shown, NMO achieves the best performance in three physics metrics. It is demonstrated that NMO learns the physical property of mass conservation in the Shallow-Water equations scenario, explaining why NMO performs well in long-term predictions. For the Rayleigh-Bénard convection scenario, the energy spectrum of NMO closely matches the target. Specifically, in terms of the

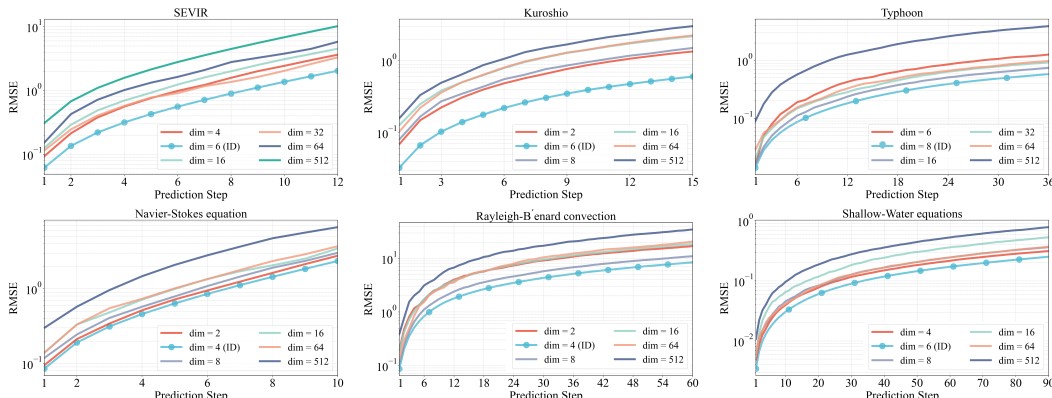

Figure 5: The prediction performance of various dimensions of the time evolution operator. The dotted lines represent the intrinsic dimension in each scenario.

Table 2: Ablation study for different dimensions of time evolution operator. Three types implementation of time evolution operator implementations including Multi-Layer Perceptron (MLP), Convolutional Neural Network(CNN) and Transformer (FORMER) structure are evaluated in the ablation experiments. The budget column shows the optimal result for model parameters (PARAM), floating point operations (FLOPs), and training time (TIME) in each dimension. The underline indicates the most accurate result in each implementation. The bold font indicates the intrinsic dimension of each scenario.

| DIMENSION | KUROSHIO | | | BUDGET | | | NAVIER STOKES | | | BUDGET | | |
|---|---|---|---|---|---|---|---|---|---|---|---|---|
| | MLP | CNN | FORMER | PARAM | FLOPs | TIME | MLP | CNN | FORMER | PARAM | FLOPs | TIME |
| 2 | 0.0430 | 0.0466 | 0.0443 | 1.5414 | 4.9180 | 49.3971 | 0.2557 | 0.2631 | 0.2631 | 5.2712 | 2.4427 | 8.0935 |
| 4 | 0.0429 | 0.0469 | 0.0461 | 1.6922 | 5.0717 | 55.4782 | **0.2547** | **0.2487** | **0.2498** | 5.5496 | 2.5147 | 8.4504 |
| 6 | **0.0421** | **0.0404** | **0.0427** | 1.8568 | 5.2375 | 64.8712 | 0.2605 | 0.2593 | 0.2528 | 5.8418 | 2.5898 | 9.6932 |
| 8 | 0.0436 | 0.0471 | 0.0482 | 2.0353 | 5.4154 | 68.3761 | 0.2607 | 0.2637 | 0.2566 | 6.1478 | 2.6678 | 9.8732 |
| 16 | 0.0431 | 0.0484 | 0.0495 | 2.8870 | 6.2474 | 78.6860 | 0.2638 | 0.2677 | 0.2607 | 7.5098 | 3.0102 | 9.9510 |
| 32 | 0.0442 | 0.0511 | 0.0524 | 3.9423 | 7.4721 | 98.3267 | 0.3021 | 0.2655 | 0.2799 | 8.3212 | 3.6217 | 10.2132 |
| 64 | 0.0477 | 0.0490 | 0.0497 | 5.7864 | 8.4464 | 111.8921 | 0.2972 | 0.2802 | 0.3021 | 9.1213 | 4.2173 | 13.3530 |

physical property of zero absolute divergence, the performance of NMO is even better than TF-Net with explicitly hard constraints.

## 4.4 ABLATION STUDY

To demonstrate the intrinsic dimension calculated by our paradigm is the optimal dimension of the latent space, we set several experiments on various dimensions for three types of time evolution operator implementations. As Figure 5 and Table 2 show, when the latent space is projected onto the calculated intrinsic dimension to learn the underlying operators, all three NMO implementations consistently achieve the best performance across all scenarios.

## 5 CONCLUSION

In this paper, we design a new operator learning paradigm with three implementations for learning the evolution of physical dynamics in intrinsic dimension. Incorporating the manifold learning algorithm, our paper mathematically and experimentally answers how to parameterize infinite-dimensional operators by a finite-dimensional parameter space. In the future, we will further explore the generalization and physics-preserving capability of our paradigm and its multi-disciplinary applications.

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

# A DETAILS FOR NEURAL MANIFOLD OPERATOR

## A.1 MODEL DETAILS

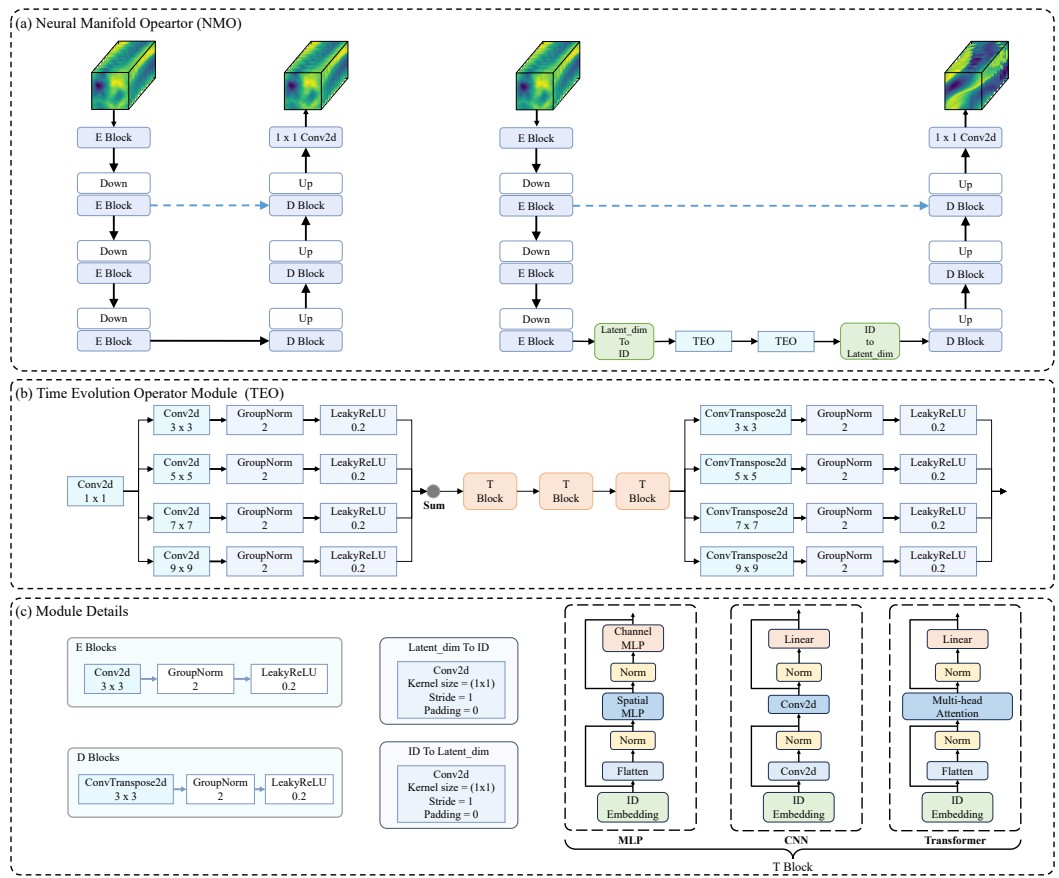

Figure 6: The architectural details of NMO.

Figure 6 shows the detailed structure of NMO. Figure 6(a) shows the overall network structure of NMO in the pretraining and training process. In the pretraining process, the model structure consists of an encoder and a decoder. In the training process, the model structure consists of an encoder, a decoder, and a time evolution operator. Figure 6(c) depicts the design of the E Block, which primarily consists of 3x3 Conv2d convolution kernels, complemented by GroupNorm for normalization and LeakyReLU as the activation function. The core function of the E Block is to downsample while elevating the feature dimension of the observed data. In contrast, the details of the D Block, also showcased in Figure 6(c), focus on upsampling and restoring latent features to the target dimension. The D Block is mainly composed of 3x3 ConvTranspose2d, GroupNorm, and LeakyReLU.

There are two linear projecting layers in the training process of the model, which aim to project the latent space into the intrinsic dimension or from the intrinsic dimension into the original latent space. The aim of the design for linear projecting layers is to learn the underlying operators by the Time Evolution Operator (TEO) module in the intrinsic dimension.

Figure 6(b) illustrates the design of the TEO, which is similar to U-Net. Initially, downsampling is performed using multiscale convolutions with kernel sizes of $\{3, 5, 7, 11\}$. Subsequently, the data passes through the T Block and is then upsampled using multiscale transpose convolution layers. The details of the three implementations based on MLP, CNN, and Transformer for T Block are shown in Figure 6(c).

## A.2 ALGORITHM DETAILS

---

**Algorithm 1** Neural Manifold Operator

---

1: **Problem Formulation:**
**Require:** Physics variable $X$ such that $X \in D$ where $D \subset \mathbb{R}^d$
**Ensure:** The physical dynamics represent the temporal evolution of $X$
2: Let the dynamical system be represented as $\frac{dX}{dt} = f(X)$
3: The discrete forms is $X_{t+\varepsilon} = \mathbf{F}(X_t, t)$
4: Aim: Create a parameterized structure $G_\theta$, to approximate the flow mapping of $X$ based on finite data.
5: **Architecture Formulation:**
6: Components:
   - Encoder $\mathcal{P}$: Maps from $\mathbb{R}^d$ to $\mathbb{R}^{d_L}$
   - Decoder $\mathcal{Q}$: Maps from $\mathbb{R}^{d_L}$ to $\mathbb{R}^d$
   - Time Evolution Operator $\mathcal{K}$: Acts on $v_t$ in $\mathbb{R}^{d_L}$ producing an evolved state
7: In the pretraining process: Encoder and Decoder are self-supervised trained by reconstruction constraint $L_{pred}$
8: When the reconstruction training reaches convergence, projecting $X$ to a latent variable $V$ in a higher-dimensional latent space $\mathbb{R}^{d_L}$
9: Computing the minimum dimensional submanifold representation of the latent variable $V$ and its dimension $m$ is considered as the intrinsic dimension $d_{\text{id}}$
10: Linear projecting $\mathcal{L}$ from the latent space to the compact space
11: Compose the architecture as: $G_\theta = \mathcal{Q} \circ \mathcal{L}^{-1} \circ \mathcal{K}_{\text{id}} \circ \mathcal{L} \circ \mathcal{P}$
12: **Projection Network in Latent Space:**
13: Objective: Facilitate mappings between the physical and latent spaces
14: Components: Encoder $\mathcal{P}$ and Decoder $\mathcal{Q}$
15: Optimize by minimizing the reconstruction loss: $L_{rec} = ||\mathcal{Q} \circ \mathcal{P}(X_t) - X_t||_2^2$
16: Aim: Determine the minimal dimension representation of the manifold $\mathcal{M}$
17: Calculate the intrinsic dimension $d_{\text{id}}$ by maximum likelihood estimation.
18: **Formulation of the Time Evolution Operator:**
19: Given the intrinsic dimension $d_{\text{id}}$, linearly project $\mathcal{V}_t$ to $\mathcal{W}_t$
20: Decompose $\mathcal{K}_{\text{id}}$ into: $\mathcal{K}_u \circ \mathcal{K}_e \circ \mathcal{K}_d$
21: Implement the evolution capture operator $\mathcal{K}_e$ using MLP, CNN, or Transformer
22: Objective: Minimize prediction loss $L_{pred} = ||G_\theta(X_t) - X_{t+\varepsilon}||_2^2$

---

---

**Algorithm 2** Intrinsic Dimension Calculation

---

1: **Aims:** Calculating the minimal manifold representation of the latent variable $\mathcal{V}_t$ to approximately calculate the intrinsic dimension $d_{\text{id}}$.
2: **Nearest Neighbor Computation:**
3: **function** KNN($\mathcal{V}_t$, n_neighbors, n_jobs)
4:     Convert $\mathcal{V}_t$ to a Numpy array and move to CPU
5:     Initialize a nearest neighbor object with given neighbors and jobs
6:     **return** distances and indices of nearest neighbors
7: **end function**
8: **Calculate Intrinsic Dimension by Maximum Likelihood Estimate:**
9: **function** MLE($\mathcal{V}_t$, dists, $k$)
10:     Compute matrix $A$ using distances
11:     Compute intrinsic dimension from $A$
12:     **return** the expected value of dimension $m$
13: **end function**
14: **Algorithm Flow:**
15: **function** ESTIMATE_DIMENSION(latent_embedding, k)
16:     Extract shape of latent_embedding
17:     Reshape and rearrange latent_embedding for KNN
18:     Get distances using KNN
19:     Maximum Likelihood Estimate for calculating intrinsic dimension
20:     **return** estimated dimension $m$
21: **end function**

---

# B  DETAILS FOR BENCHMARKS AND EXPERIMENT

We have summarized benchmark configurations in Table 3. All experiments are conducted on a single NVIDIA A100 40GB GPU.

Table 3: Detailed information for benchmarks. $N_{tr}$, $N_{ev}$ and $N_{te}$ represent the number of instances in the training, evaluation, and test sets, while $I_l$ and $O_l$ denote the lengths of the input and prediction sequences, respectively. $N_v$ denotes the number of variables.

| Dataset | $N_{tr}$ | $N_{ev}$ | $N_{te}$ | $N_v$ | Resolution | $I_l$ | $O_l$ | Interval |
|---|---|---|---|---|---|---|---|---|
| SEVIR | 35,718 | 4465 | 4465 | 1 | (192, 192) | 13 | 12 | 5 mins |
| Kuroshio | 1660 | 208 | 208 | 2 | (128, 128) | 5 | 15 | 1 day |
| Typhoon | 4158 | 445 | 500 | 3 | (512, 512) | 6 | 36 | 1 hour |
| Navier-Stokes equation | 1000 | 100 | 100 | 1 | (64, 64) | 10 | 10 | 1 step |
| Shallow-Water equations | 1000 | 100 | 100 | 1 | (128, 128) | 10 | 90 | 1 step |
| Rayleigh-Bénard convection | 1544 | 193 | 193 | 2 | (64, 448) | 10 | 60 | 1 step |
| Diffusion-Reaction equation | 1000 | 100 | 100 | 2 | (128, 128) | 50 | 50 | 1 step |

## B.1  REAL-WORLD SCENARIOS

### B.1.1  SEVIR

**Data Description.**  SEVIR dataset is a standard benchmark that includes various types of temporally and spatially aligned image sequences for weather radar and satellite. The publicly available dataset has attracted widespread attention from the weather and climate research community. We choose Infrared Satellite imagery by the sensor named GOES-16 C09 with 6.9 $\mu$m infrared channels. The physical variable inverted by satellites in the particular wavelength range is mid-level water vapor, which is highly correlated with precipitation.

**Experiment Settings.**  In the SEVIR scenario, the configuration of NMO consists of a 4-layer encoder, a 4-layer decoder, and a 6-layer time evolution operator. In the pretraining process, we set 256 as the dimension of latent space and employ an early stopping strategy. The pretraining process stops when the reconstruction error converges to 0.002786. The intrinsic dimension is 6. In the training process, the training epoch is 100. The parameters of the encoder and decoder are frozen. We set MSE Loss and Adam optimizer with a learning rate of 0.001 in the pretraining process and training process. The time steps for the input and output tensors are 13 and 12, respectively. Each time step interval is 5 minutes. The maximum prediction length through a single forward process of the model is 1 hour.

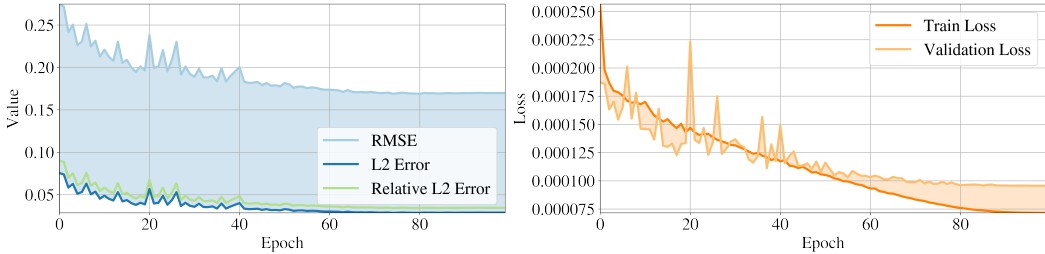

Figure 7: Training details for the SEVIR scenario. The left picture shows the variation of RMSE, L2 error and relative L2 error with iterations. The right picture shows the change in MSE loss for both the training and validation dataset with respect to iterations.

**Experiment Result.**  In the SEVIR scenario, the training details of the NMO are shown in Figure. 7. In the training process, NMO typically converges after approximately 40 epochs, reaching an L2 error of 0.02885460 and a Relative L2 error of 0.03457794. Following the settings mentioned above, we utilize the officially recommended visualization library for visual presentation. As depicted in Figure 8, the first row displays the initial conditions, comprising the 65-minute duration of historical

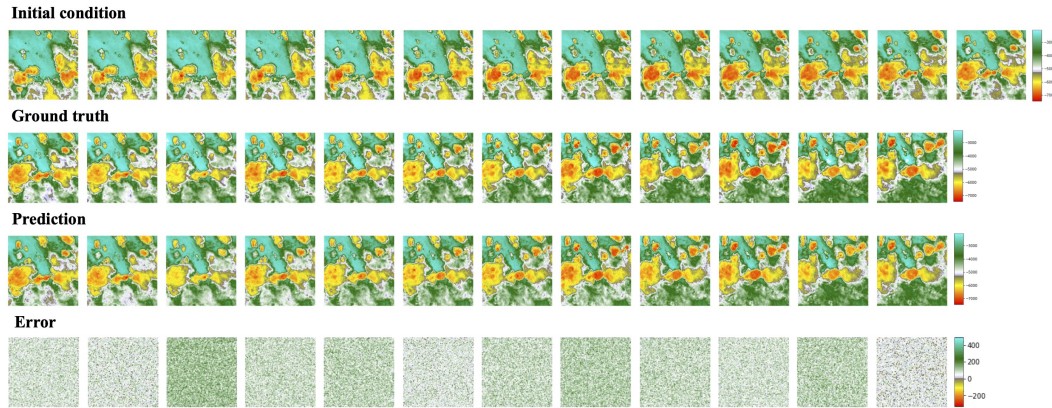

Figure 8: Visualization of the SEVIR scenario.

observation data. The second row showcases the 60-minute duration of ground truth data, while the third row illustrates the prediction results of NMO for the future 60 minutes. The fourth row visualizes the prediction error between the predictions and the ground truth.

### B.1.2 KUROSHIO

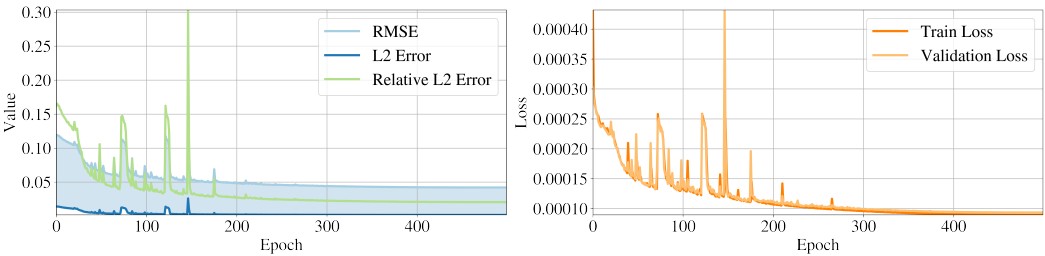

Figure 9: Training details for the Kuroshio scenario. The left picture shows the variation of RMSE, L2 error, and relative L2 error with iterations. The right picture shows the change in MSE loss for both the training and validation dataset with respect to iterations.

**Data Description.** We utilize the sea surface stream velocity data from the Copernicus Marine Environment Monitoring Service (CMEMS), which is a daily global satellite sea level product with a resolution of 0.25×0.25 degrees. Specifically, we select data from the Kuroshio region (10-42°N, 123-155°E) and the Gulf Stream region (20-52°N, 33-65°W).

**Experiment Settings.** In the Kuroshio scenario, the configuration of NMO consists of a 4-layer encoder, a 4-layer decoder, and an 8-layer time evolution operator. During the pretraining process, we set the dimension of the latent space to 256 and adopt an early stopping strategy. The pretraining process halts when the reconstruction error converges to 0.0000032. The intrinsic dimension is 6. During the training process, we set the batch size to 15 and the total epochs to 500. The parameters of the encoder and decoder are frozen. In both the pretraining and training processes, we employ a mean squared error (MSE) loss and an Adam optimizer with a learning rate of 0.001. The time steps for the input and output tensors are 5 and 15, respectively. Each time step interval is 1 day. The model's maximum prediction length through a single forward process is 15 days.

**Experiment Result.** In the Kuroshio scenario, the training details of the NMO are shown in Figure. 9. In the training process, NMO typically converges after approximately 200 epochs, reaching an L2 error of 0.00177859 and a Relative L2 error of 0.02066821.

In Figure 10, we use the historical 5-day ocean current velocity field as input and forecast the future state for the next 15 days. Surprisingly, even on the 15th day, the prediction aligns closely with the

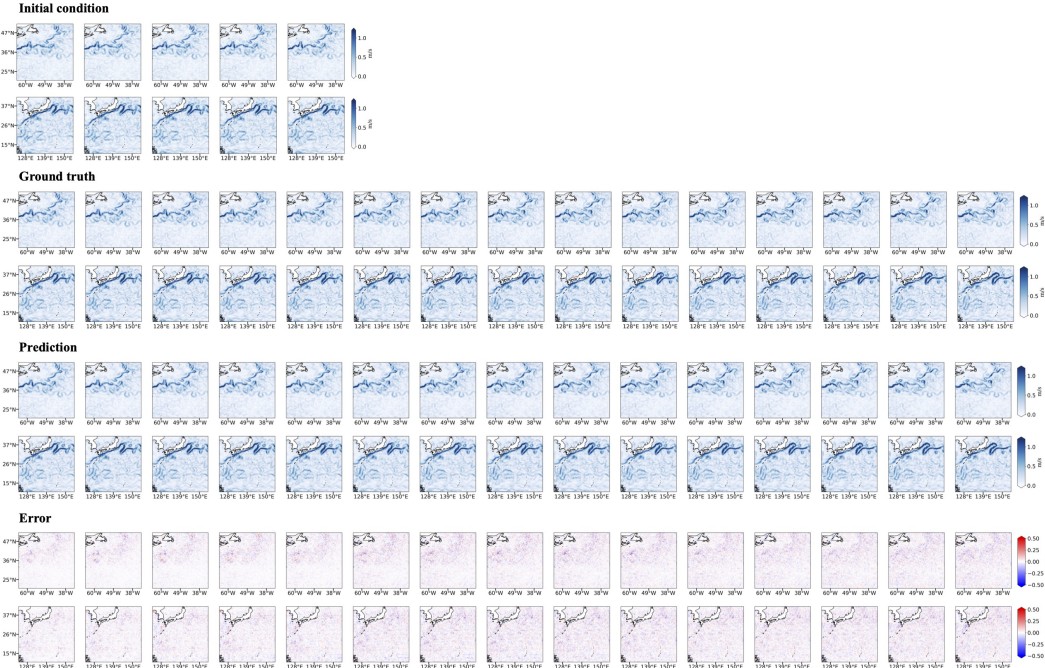

Figure 10: Visualization of the Kuroshio scenario.

ground truth in spatial patterns. Furthermore, the forecast error exhibits a subtle increase in the lead time.

### B.1.3 TYPHOON

**Data Description.** Typhoon data is obtained by the Japanese Himawari-8 Geostationary Satellite Data including three water vapor channels at high, mid, and low altitudes in the East Asia to Southeast Asia Pacific coastal region. The dataset captures the development and growth stages of typhoons through water vapor information. The time series of past meteorological satellite data contained in the dataset can be used to train the model to predict the details of typhoon development in the next 36 hours, including its position, intensity, and water vapor distribution.

**Experiment Settings.** In the Typhoon scenario, the configuration of NMO consists of a 6-layer encoder, a 6-layer decoder, and a 16-layer time evolution operator. During the pretraining process, we set the dimension of the latent space to 768 and adopt an early stopping strategy. The pretraining process halts when the reconstruction error converges to 0.0001232. The intrinsic dimension is 8. During the training process, there are 300 epochs, and we set the batch size to 1. The parameters of

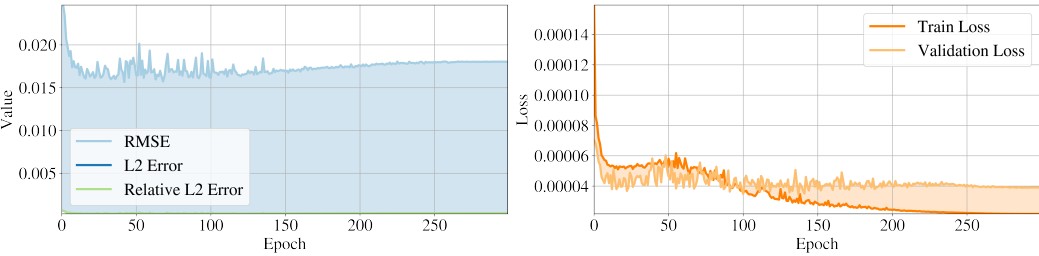

Figure 11: Training details for the Typhoon scenario. The left picture shows the variation of RMSE, L2 error, and relative L2 error with iterations. The right picture shows the change in MSE loss for both the training and validation dataset with respect to iterations.

the encoder and decoder are frozen. In both the pretraining and training processes, we employ a mean squared error (MSE) loss and an Adam optimizer with a learning rate of 0.01. The time steps for the input and output tensors are 6 and 36, respectively. Each time step interval is 1 hour. The model's maximum prediction length through a single forward process is 36 hours.

**Experiment Result.** In the context of the Typhoon scenario, the training details of NMO are depicted in Figure 11. Throughout the training, NMO typically converges after approximately 150 epochs, achieving an L2 error of 0.00032553 and a relative L2 error of 0.00032245. The visualization results are shown in Figure 12. The first section of the visualization presents the initial condition. The three rows therein represent the data from the high, medium, and low water vapor channels respectively. The subsequent 6 images depict input data over a continuous 6-hour period. In the second section, we showcase the ground reality data for the upcoming 36-hour. However, for the sake of brevity, we chose to display one image every six hours. The third section reveals NMO's predictions for the next 36 hours. The fourth row contrasts the predicted data with the actual results, providing an intuitive assessment of the prediction error.

## B.2 EQUATION-GOVERNED SCENARIOS

### B.2.1 NAVIER-STOKES EQUATION

**Data Description.** The dataset is calculated through a pseudospectral method to solve a viscous, incompressible 2-d Navier-Stokes equation with vorticity form expressed as

$$\begin{aligned}
\partial_t w(x,t) + u(x,t) \cdot \nabla w(x,t) &= \nu \Delta w(x,t) + f(x), & x &\in (0,1)^2, t \in (0,T] \\
\nabla \cdot u(x,t) &= 0, & x &\in (0,1)^2, t \in [0,T] \\
w(x,0) &= w_0(x), & x &\in (0,1)^2.
\end{aligned}$$

The forcing term is fixed as $f(x) = 0.1 \left( \sin \left( 2\pi \left( x_1 + x_2 \right) \right) + \cos \left( 2\pi \left( x_1 + x_2 \right) \right) \right)$. In order to obtain a set of different solutions for training the mapping between Banach space, the initial condition is generated by the $w_0 \sim \mu$ with $\mu = \mathcal{N} \left( 0, 7^{3/2}(-\Delta + 49I)^{-2.5} \right)$ and the boundary condition is periodic boundary conditions. We set the viscosity coefficient as $\nu = 10^{-5}$ to make the solutions become chaotic enough with time evolution.

**Experiment Settings.** In the Navier-Stokes equation scenario, the configuration of NMO consists of a 4-layer encoder, a 4-layer decoder, and an 8-layer time evolution operator. During the pretraining process, we set the dimension of the latent space to 256 and adopt an early stopping strategy. The pretraining process halts when the reconstruction error converges to 0.0015347. The intrinsic dimension is 4. During the training process, there are 100 epochs, and we set the batch size to 20. The parameters of the encoder and decoder are frozen. In both the pretraining and training processes, we employ a mean squared error (MSE) loss and an Adam optimizer with a learning rate of 0.01. The time steps for the input and output tensors are 10 and 10, respectively. The model's maximum prediction length through a single forward process is 10 time steps.

**Experiment Result.** In the context of the Navier-Stokes equation scenario, the training details of NMO are outlined in Figure 13. Throughout its training, NMO typically converges after roughly 70 epochs, achieving an L2 error of 0.05058351 and a relative L2 error of 0.03410483. The visualization results are presented in Figure 14. The first row depicts the initial conditions with 10 time steps. The second row showcases the ground truth data with 10-time steps, while the third row illustrates NMO's predictions for the subsequent 10 time steps. The fourth row offers a visual comparison between the predicted results and the actual data, highlighting the prediction error.

### B.2.2 SHALLOW-WATER EQUATIONS

**Data Description.** The dataset is calculated by a comprehensive finite volume solver to solve 2-d Shallow-Water equations for free-surface flow, expressed as

$$\partial_t h + \partial_x h u_x + \partial_y h u_y = 0$$

$$\partial_t h u_x + \partial_x \left( u_x^2 h + \frac{1}{2} g_r h^2 \right) = -g_r h \partial_x b$$

$$\partial_t h u_y + \partial_y \left( u_y^2 h + \frac{1}{2} g_r h^2 \right) = -g_r h \partial_y b$$

where $h$ is the depth of the water column, $u_x$ and $u_y$ is the component of velocity with respect to the x-axis and y-axis, $b$ is the bathymetry variation and $g_r$ is the gravitational acceleration.

**Experiment Settings.**    In the Shallow-Water equations scenario, the configuration of NMO consists of a 3-layer encoder, a 3-layer decoder, and a 6-layer time evolution operator. During the pretraining process, we set the dimension of the latent space to 256 and adopt an early stopping strategy. The pretraining process halts when the reconstruction error converges to 0.0000312. The intrinsic dimension is 6. During the training process, there are 100 epochs, and we set the batch size to 10. The parameters of the encoder and decoder are frozen. In both the pretraining and training processes, we employ a mean squared error (MSE) loss and an Adam optimizer with a learning rate of 0.01. The time steps for the input and output tensors are 10 and 90, respectively. The model's maximum prediction length through a single forward process is 90 time steps.

**Experiment Result.**    In the context of the Shallow-Water equations scenario, the training details of NMO are outlined in Figure 15. Throughout its training, NMO typically converges after roughly 25 epochs, achieving an L2 error of 0.0000079 and a relative L2 error of 0.00270999. The visualization results are presented in Figure 16. The first row depicts the initial conditions with an input of 10-time steps. The second row showcases the ground truth data for those 90-time steps, while the third row illustrates NMO's predictions for the subsequent 90-time steps. The fourth row offers a visual comparison between the predicted results and the actual data, highlighting the prediction error.

### B.2.3    RAYLEIGH-BÉNARD CONVECTION

**Data Description.**    The dataset is calculated by the Lattice Boltzmann Method to solve the 2-d fluid thermodynamics equations for two-dimensional turbulent flow, and its general form is expressed as

$$\nabla \cdot \mathbf{u} = 0$$

$$\frac{\partial \mathbf{u}}{\partial t} + (\mathbf{u} \cdot \nabla)\mathbf{u} = -\frac{1}{\rho_0} \nabla p + v \triangle \mathbf{u} + [1 - \alpha (\mathrm{T} - \mathrm{T}_0)] \mathbf{X}$$

$$\frac{\partial \mathrm{T}}{\partial \mathrm{t}} + (\mathbf{u} \cdot \nabla)\mathrm{T} = \kappa \triangle \mathrm{T}$$

where $g$ is the gravitational acceleration, $\mathbf{X}$ is the acceleration due to the body-force of the fluid parcel, $\rho_0$ is the relative density, $T$ denotes temperature, $T_0$ is the average temperature, $\alpha$ denotes the coefficient of thermal expansion and $\kappa$ denotes the coefficient of thermal conductivity. The simulation parameters of the data respectively are Prandtl number = 0.71, Rayleigh number = $2.5 \times 108$, and the maximum Mach number = 0.1.

**Experiment Settings.**    In the Rayleigh-Bénard convection scenario, the configuration of NMO consists of a 2-layer encoder, a 2-layer decoder, and a 4-layer time evolution operator. During the pretraining process, we set the dimension of the latent space to 512 and adopt an early stopping strategy. The pretraining process halts when the reconstruction error converges to 0.0129318. The intrinsic dimension is 4. During the training process, there are 300 epochs, and we set the batch size to 1. The parameters of the encoder and decoder are frozen. In both the pretraining and training processes, we employ a mean squared error (MSE) loss and an Adam optimizer with a learning rate of 0.01. The time steps for the input and output tensors are 10 and 60, respectively. The model's maximum prediction length through a single forward process is 60 time steps.

**Experiment Result.**    In the Rayleigh-Bénard convection scenario, Figure 17 depicts the training details of the NMO. Throughout the training process, the NMO typically converges after about 150 cycles and reaches an L2 error of 0.0201180, while the relative L2 error is 0.0053570. As Fig.

18 shown, the first column displays the 10 time-steps for the initial condition; the second column illustrates the ground truth, which only shows time-step indexes of 6, 12, 18, ..., 54, 60; the third column shows the NMO prediction results; and the fourth column presents the error visualization.

### B.2.4 DIFFUSION-REACTION EQUATION

**Data Description.** The dataset is calculated by the finite volume method for spatial discretization and the fourth-order Runge-Kutta method for time integration to solve a 2-d diffusion-reaction equation expressed as

$$\partial_t u = D_u \partial_{xx} u + D_u \partial_{yy} u + R_u,$$
$$\partial_t v = D_v \partial_{xx} v + D_v \partial_{yy} v + R_v,$$

where $D_u$, $D_v$, $R_u$, and $R_v$ are the diffusion coefficient and reaction function for the activator and inhibitor, respectively. The reaction functions are defined as

$$R_u(u,v) = u - u^3 - k - v$$
$$R_v(u,v) = u - v$$

where constant number $k = 5 \times 10^{-3}$, the diffusion coefficients $D_u = 1 \times 10^{-3}$ and $D_v = 5 \times 10^{-3}$.

**Experiment Settings.** In the Diffusion-Reaction equation scenario, the configuration of NMO consists of a 4-layer encoder, a 4-layer decoder, and an 8-layer time evolution operator. During the pretraining process, we set the dimension of the latent space to 256 and adopt an early stopping strategy. The pretraining process halts when the reconstruction error converges to 0.00000292. The intrinsic dimension is 6. During the training process, there are 100 epochs, and we set the batch size to 2. The parameters of the encoder and decoder are frozen. In both the pretraining and training processes, we employ a mean squared error (MSE) loss and an Adam optimizer with a learning rate of 0.01. The time steps for the input and output tensors are 50 and 50, respectively. The model's maximum prediction length through a single forward process is 50 time steps.

**Experiment Result.** In the context of the Diffusion-Reaction equation, the training process of NMO is depicted in Figure 19. Throughout the training, NMO typically converges around 400 epochs, achieving an L2 error of 0.00000063 and a relative L2 error of 0.00005421. The associated visualization results are shown in Figure 20. Given that the equation involves two variables, the visualization is split into two sections, left and right. On the left, the initial conditions are presented, followed by the ground truth values. Subsequently, the NMO's predictions are displayed, and the last section presents the prediction error.

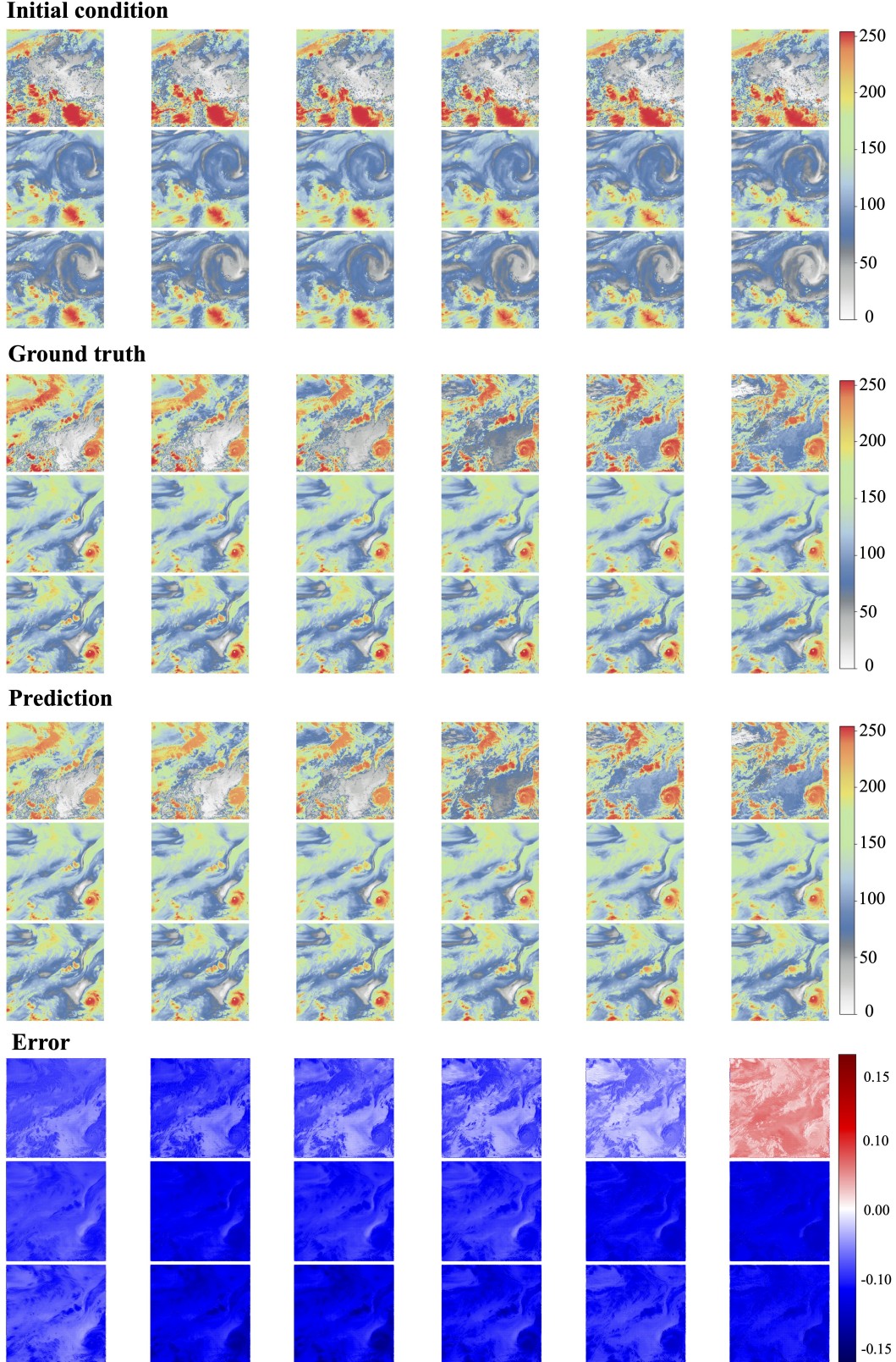

Figure 12: Visualization of the Typhoon scenario.

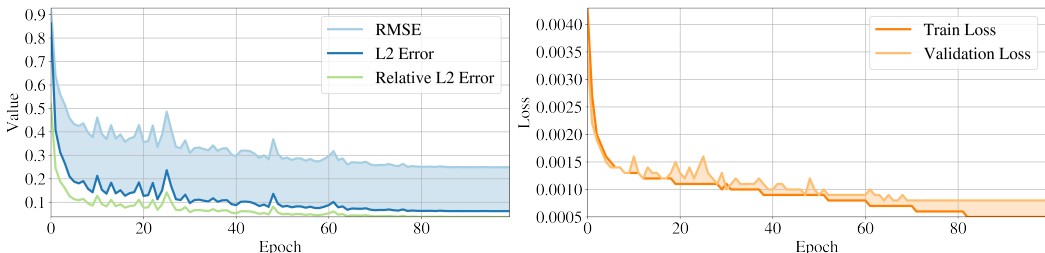

Figure 13: Training details for the Navier-Stokes equation scenario. The left picture shows the variation of RMSE, L2 error and relative L2 error with iterations. The right picture shows the change in MSE loss for both the training and validation dataset with respect to iterations.

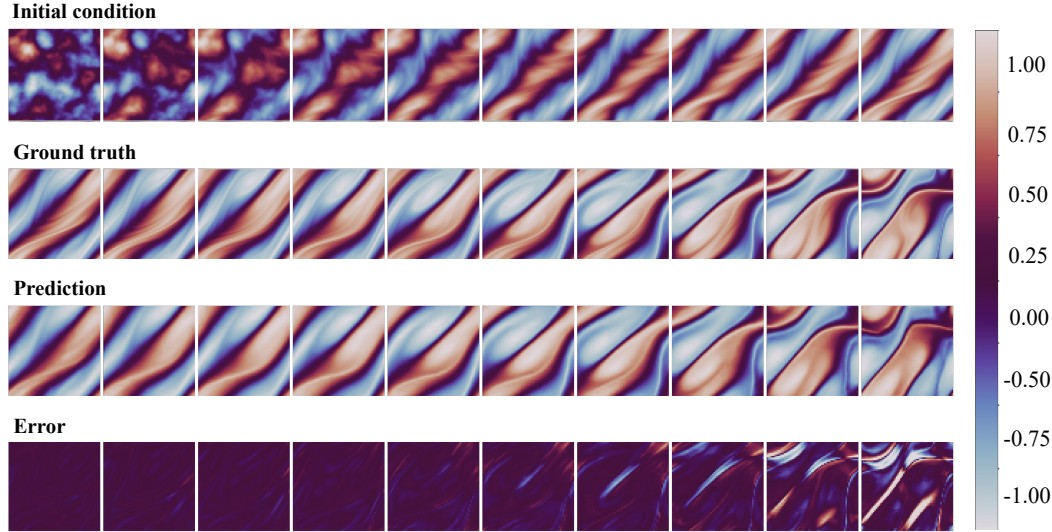

Figure 14: Visualization of the Navier-Stokes equation scenario.

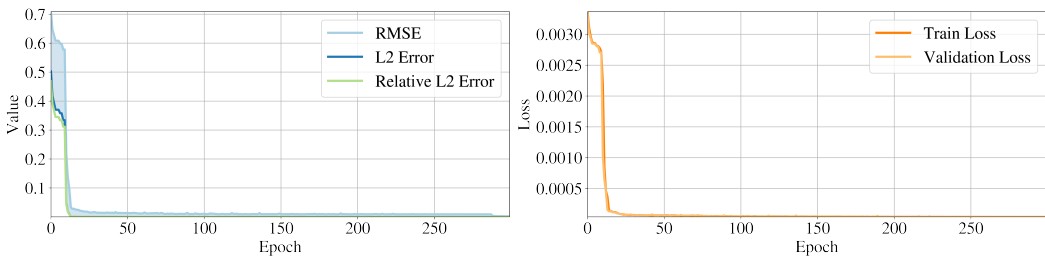

Figure 15: Training details for the Shallow-Water equations scenario. The left picture shows the variation of RMSE, L2 error and relative L2 error with iterations. The right picture shows the change in MSE loss for both the training and validation dataset with respect to iterations.

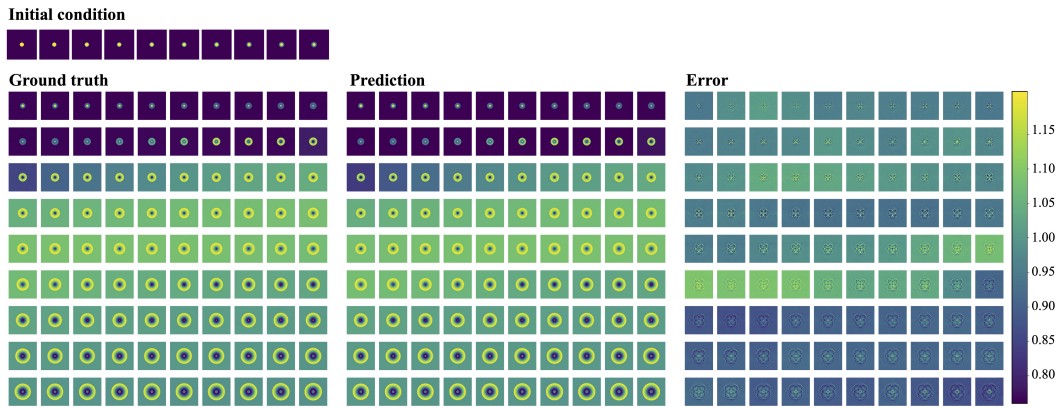

Figure 16: Visualization of the Shallow-Water equations scenario.

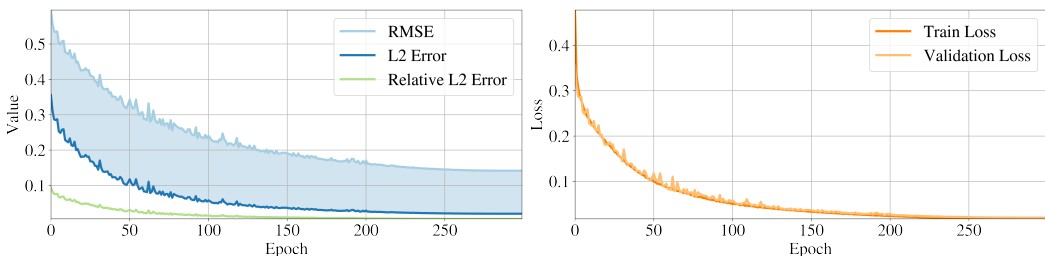

Figure 17: Training details for the Rayleigh-Bénard convection scenario. The left picture shows the variation of RMSE, L2 error and relative L2 error with iterations. The right picture shows the change in MSE loss for both the training and validation dataset with respect to iterations.

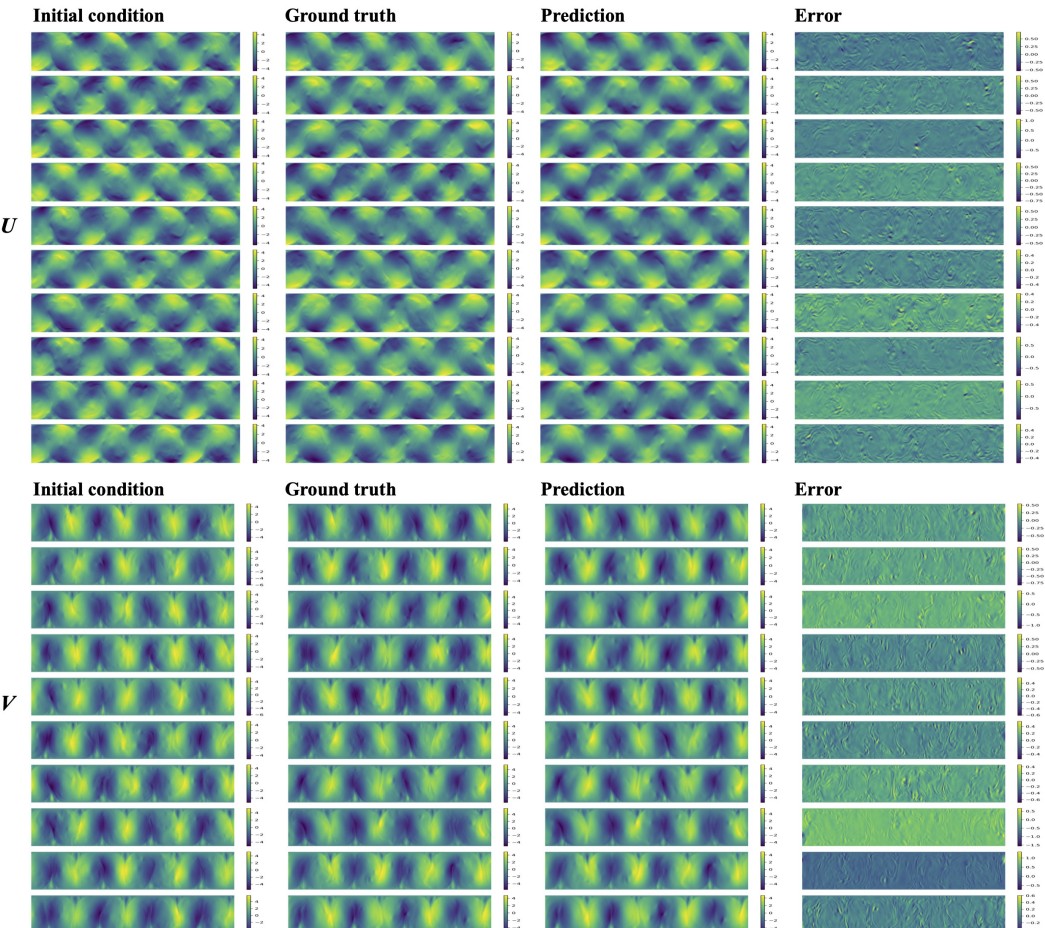

Figure 18: Visualization of the Rayleigh-Bénard convection scenario.

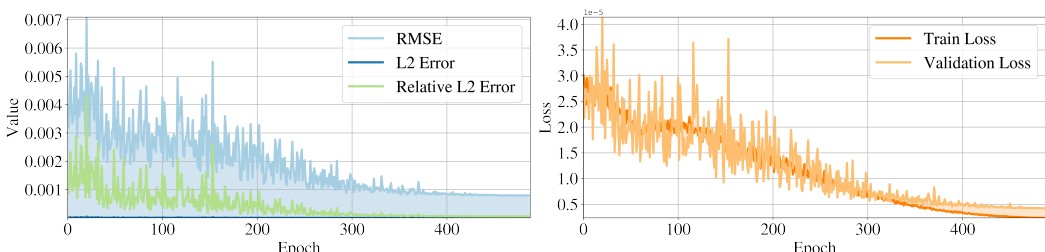

Figure 19: Training details for the Diffusion-Reaction equation scenario. The left picture shows the variation of RMSE, L2 error and relative L2 error with iterations. The right picture shows the change in MSE loss for both the training and validation dataset with respect to iterations.

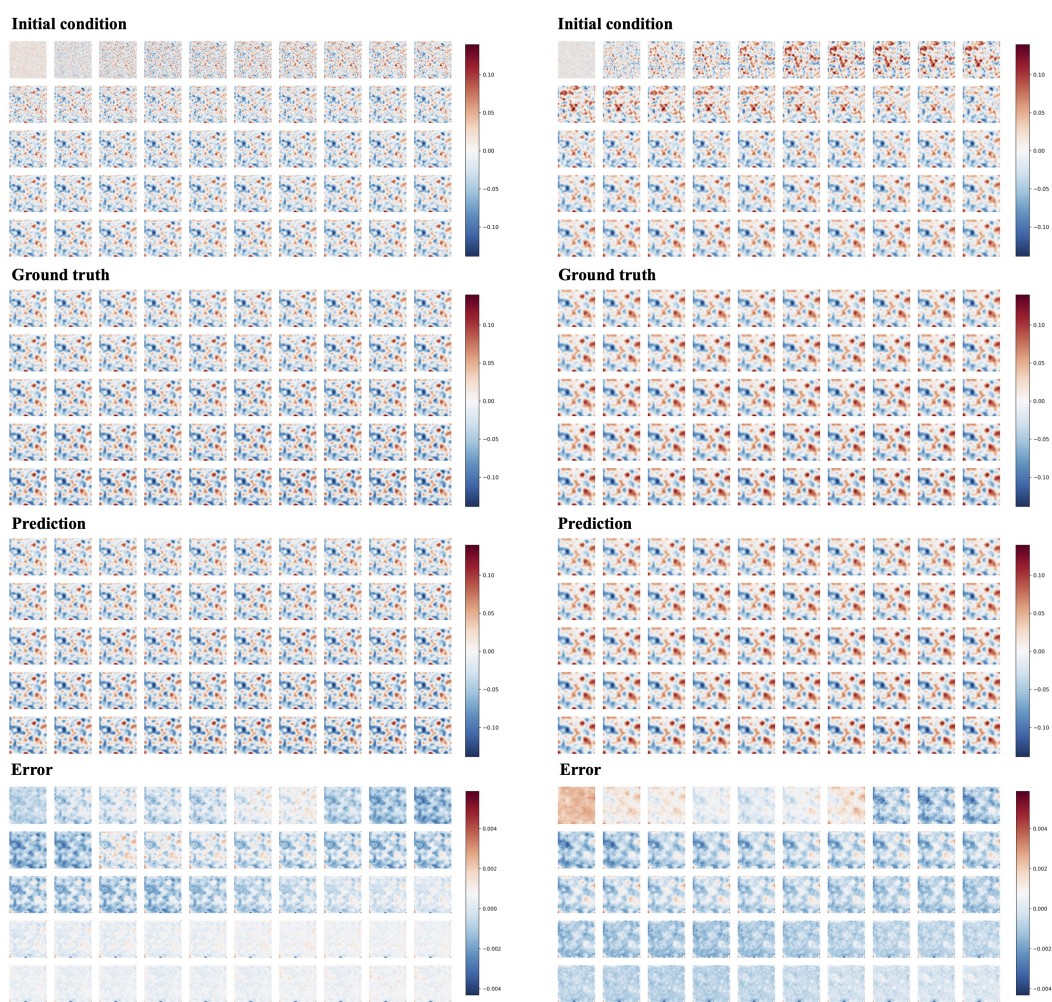

Figure 20: Visualization of the Diffusion-Reaction equation scenario

