# OpenReview forum: "Neural Manifold Operators for Learning the Evolution of Physical Dynamics"
_ICLR.cc/2024/Conference — ICLR 2024 Conference Withdrawn Submission_

### Official Review · Reviewer_bPYN · 2023-10-27

**Soundness:** 4 excellent
**Presentation:** 3 good
**Contribution:** 4 excellent
**Rating:** 8
**Confidence:** 2

**Summary:**

The paper introduces an operator learning framework that includes a projection onto a lower-dimensional latent manifold. The authors show state-of-the-art results on many real world datasets and simulations across multiple models from vision, forecasting, PDE nets and neural operator litrature.

**Strengths:**

This seems like a very strong paper with excellent results on a wide variety of datasets. The overarching idea is well motivated and explained, and a lot of the details are given in the supplementary I particularly like that the authors included a large mixture of real world datasets and equation based simulations and showed improvement over models from various fields.

**Weaknesses:**

I think this is an excellent paper and only has some minor points when it comes to presentation.

1) Section 3.3.2 seems to be central to key innovation of the paper, which introduces a lower-dimensional manifold into an operator learning framework. Nonetheless, it is not presented or explained well and to get the central point the reader has to go seek out Costa et al. 2005. I would suggested adding an illustration and some more explanations to these ideas which are central to the paper's claims.

2) All of the figure captions are very lacking:
   Fig 1. is not referred to in the text and has very small text that is hard to read or interpret. Furthermore, it would be good if the caption could be elaborated to include a walk-through of the figure. For example there is no reference to (a), (b), (c) etc..
  Fig 2., Fig 3. There is almost no caption. Again, it would be useful to have an informative caption across the paper.
 Fig 4. Very packed and not explained well. What is the prediction step showing? How far into the future is this predicting? What are the initial conditions, etc.? Also, some plots have many colors, and one has to fish out the "target" and compare it to the different models by eye.

3) I think this paper could use a good read-through for grammatical errors. examples:
Section 3.3.2 "There exists that the latent variables ...", "intrinsic dimension in local as the following..." and more.

**Questions:**

There are some critical points in the text that were hard to follow. In particular Sec 3.3.2 lacks clarity. Some of the questions are:

1.  Is the intrinsic dimension differentiable? Is it computed in pre-training and then frozen?
2. How do you choose "k" (Eqs 9-10) and what is the effect of "k" on the results?
3. Eq (5) --> can you clearly mark what is trainable? Q, P are frozen, does theta parameterizes both L and K?
4. Section 3.4. a notation suggestion --> K_id and R^id should probably be called K_m and R^m as this is an estimation of the intrinsic dimension but no guarantee to always converge to that.
5. 4.4. Ablation study: It is impossible to assess these results without error bars. I can't tell if this improvement will be washed away by statistical variations. This ablation study is not described in nearly enough detail, despite the fact that it is central to one of the paper's main claims (from page 2): "...the redundant dimension representation of the underlying operators leads to several problems...". Furthermore, I expected to see a study on how well the determined "m" (using algorithm 3.3.2) converges to the true m=id and under what conditions.

---

> ### Author Response · Authors · 2023-11-19
> **Our response for Reviewer bPYN**
>
> Dear reviewer, we very appreciate for your careful reading and kind help in reviewing our work. We are largely encouraged and have learned your suggestions carefully.
>
> **(1) Reply for the "central point is lacking enough illustrate"**.
>
> At first, we recognized that this part is not our original mathematical theory and omit some contents. In order for the completeness of mathematics, we rewrite this part to illustrate the  local neighborhood of these points by KNN algorithm and manifold learning methods.
>
> **(2) Reply for the "lacking enough illustrate for figure caption"**.
>
> Thank you for pointing out these issues. We have revised the captions accordingly, adding more details that were originally present in the main text and appendix into the figure captions. For instance, in Figure 4, we have added more information about prediction length and input length.
>
> **(3) Reply for "grammar error in Method part".**
>
> We have revised the Method part. On the one hand, we have reduced overly verbal and hand-wavy expressions, and on the other hand, we have augmented the proof of operator equivalence.
>
> **(4) Is the intrinsic dimension differentiable? Is it computed in pre-training and then frozen?**
>
> The intrinsic dimension is calculated upon completion of the pre-training. Because after the pre-training, the parameters of encoder and decoder are frozen and the latent tensor after the encoder are enable to be used for calculating the intrinsic dimension.
>
> **(5)How do we choose "k" (Eqs 9-10) and what is the effect of "k" on the results?**
>
> "k" is a hyper-parameter of our manifold algorithm. In plain language, manifold method to calculate submanifold by KNN method and define a neighborhood, where points outside this neighborhood are considered to have infinite distance, for the purpose of employing the KNN algorithm for clustering. If k is excessively small, it might lead to the "short-circuiting" of submanifolds that it should not be connected. However, according to our experiments, "k" is not so sensitive to the result and it is accustomed to calculate by the square root of the total number of points.
>
> **(6) Eq (5) --> can you clearly mark what is trainable? $Q$, $P$ are frozen, does theta parameterizes both $\mathcal{L}$ and $\mathcal{K}$?**
>
> In Eq(5), $Q$, $P$ are frozen. $\mathcal{L}$ is non-trainable linear projection. In train process, $\mathcal{K}_{\text{id}}$ is trainable by prediction loss.
>
> **(7) Section 3.4. a notation suggestion --> K_id and R^id should probably be called K_m and R^m as this is an estimation of the intrinsic dimension but no guarantee to always converge to that.**
>
> Yes, it is. We have revised the paper and change the notation system. $\mathcal{K}_{\text{id}}$ is ideal situation. $\mathcal{K}_m$ is obtained by the optimal estimation.
>
> **(8) About ablation study.**
>
> Due to the intrinsic dimension of complex dynamics and real-world dynamics cannot be analytically calculated, we have got to experimentally prove that when we set the dimension is $m$, NMO get the better performance independent network implementation. In our ablation study, enough experiments have had this effect and I think it is not a coincidence. If it is statistical variations, it should not appear that all ablation experiments show $m$ is the optimal dimension.
>
> Finally, thank you for your review for our paper. According to your suggestion, we have improved our paper furtherly.

---

### Official Review · Reviewer_MieG · 2023-10-29

**Soundness:** 1 poor
**Presentation:** 1 poor
**Contribution:** 1 poor
**Rating:** 3
**Confidence:** 3

**Summary:**

This paper proposes a method of operator learning for predicting physical phenomena in which the latent space is extracted from the data and the dynamics on the latent space is estimated. In particular, the proposed method is combined with a method for estimating the dimension of the latent space.

**Strengths:**

Numerical experiments have been conducted on many specific examples.

**Weaknesses:**

This paper is very poorly written. Symbols are not used in a uniform manner and there are also undefined symbols. They are too numerous to enumerate, but for example, in the first paragraph of section 3.2, the operator $\_mathcal{K}: \{ v_t: D_t \to \mathbb{R}^{d_L} \to v_{t+\varepsilon}: D_{t+\varepsilon} \to \mathbb{R}^{d_ L} \}$ is introduced; however the symbols $D_t$ and $D_{t+\varepsilon}$ are not explained. It can be understood that these are the domains of $v_t$ and $v_{t+\varepsilon}$, but if so, this implies that the domains are assumed to change with time. However, it is not explained how the proposed method handles the evolution of the domain. In addition, there are many grammatical errors. This is just my impression but if this paper is submitted to a major journal, this paper will be rejected by the editor without review.

The novelty of the method is also questionable. The technical contribution seems to be the estimation of the dimensions of the latent space, but this is achieved by simply applying an existing method.

**Questions:**

Numerical experiments indicate that the proposed technique would improve the performance, so I encourage the authors to carefully rewrite the paper and resubmit it.

---

> ### Author Response · Authors · 2023-11-19
> **Our response for Reviewer MieG**
>
> We greatly appreciate your time spent for reading our paper and providing valuable criticism for our Method part. We have revised our notation system and added a description of all variables in the appendix part. We rewrite the notation of the operator mapping $G$ as
>
> $ G_{\theta}: R^{n} \times P(T) \rightarrow R^{n}\times P(T), \theta \in \Theta $
>
> where $P(T)$ is the power set of $T$. In plain language, $P(T)$ contents all possible subsets of $T$. Based on the operator definition, we rewrite the whole mathematics prove. Furthermore, we provide the proof that the operator for submanifold representation is equivalent to the original high-dimensional underlying operator. Using the manifold method to do the intrinsic dimension representation for the complex physical dynamics and get the state-of-the-art performance in both accuracy and efficiency is our novelty. We supplemented the proof of operator equivalence, which is another theoretical novelty.
>
> Hopefully, our revised version can address your concerns and make our idea accessible to all readers. Apart from the above responses, you can also read our response to other reviewers for more information.

---

> > ### Comment · Reviewer_MieG · 2023-11-22
> >
> > Thank you very much for your reply. Unfortunately, the quality of the manuscript does not appear to be sufficient, as typographical errors and inconsistencies in notation seem to remain. I suppose that this paper would be more valuable if it could be carefully rewritten and resubmitted.

---

### Official Review · Reviewer_KyVY · 2023-10-31

**Soundness:** 3 good
**Presentation:** 2 fair
**Contribution:** 3 good
**Rating:** 5
**Confidence:** 3

**Summary:**

The paper studies the class of methods for learning physical dynamics based on neural operators, that is where the objective is to learn an operator (in general nonlinear) acting on an infinite-dimensional function space. To achieve this task, usual approach is to learn representation manifold with encoder-decoder architecture and an approximation operator defined on it.

Authors identify that the choice of the dimension of the representation space has profound impact on the overall performance and propose a two step procedure to properly choose it.

First, by pertaining an autoencoder (AE) with a chosen latent dimension, a representation space of the trajectory is identified and its intrinsic dimension is computed as suggested in (Levina & Bickel, 2004; Costa et al., 2005). Then, freezing the parameters of pretrained autoencoder, during the training process linear reduction to intrinsic dimension, time evolution operator and linear extension to the latent dimension are learned.

The overall architecture named Neural Manifold Operator, learns nonlinear function that evolves the states of the physical system in time and its performance is tested on various datasets and against diverse operator learning baselines.

**Strengths:**

The paper is, apart of few issues reported bellow, well written. Main contribution is clearly stated and honestly reported. Relevant literature is well documented. The class of learning algorithms that is studied as well as the problem that the paper solves is are relevant. Experimental section elaborated and results are good.

**Weaknesses:**

My overall  impression of the paper is good. However, the following issues currently limit the overall score. If those are resolved, I am ready to re-evaluate my assessment.

A) Mathematical inconsistencies and overly informal presentation of the equations:
1) In Eq. (3) when defining $G$, notation is not clear. $X_t$ is a state depending on time step $t$ while $\Theta$ is a set of parameters. I imagine it should be written $G_{\varepsilon}\colon D\times \Theta \to D$, where $\varepsilon$  is the hyperparameter of the NMO model defining $\varepsilon$-ahead nonlinear evolution of $X_t$. One should comment the the problem is time-homogeneous dynamical system, hence $G_\varepsilon$ does not depend on time.

2) To train the model it is assumed to need points and their $\varepsilon$-ahead evolution, is this correct? If yes, then in the discussion that follows Eq. (3),  $X_{t:t+\varepsilon}$ should be corrected to $X_{\varepsilon:t+\varepsilon}$.

3) In Eq. (6) notion of expectation is not clear, since there is no notion of randomness and considered time horizon is not clear. What is considered as "true-operator" here? In the way how it is defined $G$ cannot play this role? I guess that a proper way to write the risk is to use random initial point, integral over time and compare $G_\theta$ to $\mathbf{F}$ defined in Eq. (2).

3) In discussion bellow Eq. (11) the notation $\mathcal{T}_{[\cdot]}$ is explained when defining operators ${\mathcal K}\_d$, $\mathcal{K}\_e$ and $\mathcal{K}\_u$.

B) As reported in Section 3, the NMO architecture is designed to learn nonlinear evolution of deterministic dynamical system. Since according to Koopman operator theory there exists a representation space where the evolution operator, here denoted as $\mathcal{K}_{id}$ is linear. Which that in mind, it would be worth to explain rationale for ETO architecture to learn nonlinear operator. Clearly, nonlinear operator is needed in more general inverse problems, but since this proposal is tailored for learning the flow maps from trajectories, I think that this aspect should be at least commented. If possible, it would be also useful including some of the Koopman based baselines in the experiments.

C) Minor issues:
1) Since matrix $\mathcal{L}$ is not squared, does $\mathcal{L}^{-1}$ in Eq (5) denotes pseudoinverse?
2) $E$ in Eq. (10) should be expectation.
3) Line bellow Eq. (11), decomposition should read composition.

**Questions:**

-

**Details Of Ethics Concerns:**

/

---

> ### Author Response · Authors · 2023-11-16
> **Our response for Reviewer KyVY (Question A)**
>
> We greatly appreciate your time spent for reading our paper and providing valuable review. According to your review, we have revised our paper.
>
> **(1) Reply for "Mathematical inconsistencies and overly informal presentation of the equations"**
>
> Thanks for your comments about the unclear mathematical symbols. We have to admit that our mathematical descriptions are not perfect. At first, in order to make the article easy to follow, some non-original mathematical theories were omitted. Based on your review, we have revised the notation system in the paper and added explanations for all mathematical notation in the appendix.
>
> 1. $G$ is the operator mapping from the physical variable $X$ at time $t$ as input to the variable at time $t+\varepsilon$ output while $\Theta$ is the finite-dimensional parameter space. The notation is followed by the previous related articles about neural operator[1,2]. Because our model design for sequence-to-sequence prediction, which means we learn the map from a time series(i.e. t:0-9) to a time series(i.e. t:10-100). We didn't have the autoregression strategy. So our paradigm are applied in autonomous and non-autonomous system. In order to be more clear, we rewrite the notation of the operator mapping $G$ as
>    $$
>    G_{\theta}: R^{n} \times P(T) \rightarrow R^{n}\times P(T), \theta \in \Theta
>    $$
> where $P(T)$ is the power set of $T$. In plain language, $P(T)$ contents all possible subsets of $T$.
>
> 2. Our paradigm is not $\varepsilon\$-ahead evolution. More details are discussed above.
> 3. The method of calculating the dimensionality of data tensor by maximum likelihood estimation[3] is not our original work, . Our novelty is that calculating the optimal representation of operators by the method. We will supplement the proof of the equivalence of the original operator and the operator with the submanifold representation. According to "no notion of randomness",  we have mentioned that in our paper "The number of the observation points falling into the k-nearest ball $B$ can be approximated by a Poisson process" to introduce the random variable[3]. In our design, in plain language, $G$ are the mapping with objective paramter $\Theta$ in general form, $G_\theta$ denotes the operator mapping are parameterized by a neural network. We have introduced new notation $\theta^{\dagger} \in \Theta $  to describe the optimal parameters. As you said, we are trying to learn a map $G_{\theta}$ to get the time evolution, but it is not exactly equivalent to $\mathbf{F}$ in Eq. (2).
> 4. In order to improve the performance furtherly, we design a relatively complicated time evolution module, it is the reason why we introduce a mathematical description for operator decomposition in Eq. (11).
>
> [1] Neural operator: Learning maps between function spaces. *Journal of Machine Learning Research*, 2022.
>
> [2] Fourier neural operator for parametric partial differential equations. *ICLR 2021*.
>
> [3] Maximum likelihood estimation of intrinsic dimension. *Advances in neural information processing systems*, *17*, .

---

> ### Author Response · Authors · 2023-11-19
> **Our response for Reviewer KyVY (Question B and C)**
>
> **(2) About ETO and Koopman operator**
>
> Our theory follows the Green function integral theory (such as FNO). We have revised our Method part to add the mathematics illustration. It seems plausible to use Koopman theory to illustrate our paradigm becasue we use auto-encoder structure to reconstruct the physical variables, which can be regarded as learning the observation function and its inverse function. However, linear representation for the Koopman operator seems more intuitive but our ETO is nonlinear. If using the theory to illustrate, maybe we need to prove that it is invariant subspace of the Koopman operator after submanifold representation. I'm not sure if it's mathematically easy to prove, but we think it's a good idea for the further work.
>
> **(3) Reply for "Minor issues"**
>
> Thanks for pointing out some typos of our paper. We have revised our paper.
>
> When  $\mathcal{L}$ is not squared, $\mathcal{L}^{-1}$ denotes pseudoinverse.
>
> $E$ is expectation.
>
> Eq. (11) should be composition.
>
> Finally, we appreciate your valuable review for our paper and helping us to improve our work.

---

> > ### Comment · Reviewer_KyVY · 2023-11-22
> > **Reply to the authors**
> >
> > I thank the authors for their response. I have read the other reviews and the discussions that followed (as of this writing), while, unfortunately, I didn’t find the submitted revision to check how certain corrections are made.
> >
> > My comments remain only partially addressed. In particular, I still do not find the presentation of the problem mathematically consistent. Notation and setup of (Kovachki et al. 2023) is loosely followed.
> >
> > Current opinion: In this form, I don’t find the paper suitable for the acceptance, and it is not clear (as of writing) that the authors will be successful in addressing properly all the raised questions in the promised revision.

---

### Official Review · Reviewer_ivj1 · 2023-11-04

**Soundness:** 1 poor
**Presentation:** 1 poor
**Contribution:** 1 poor
**Rating:** 3
**Confidence:** 3

**Summary:**

The paper aims at modeling a solution of the physical dynamics systems on infinite dimensional spaces. To do that, the paper proposes a pipeline that consists of (1) a pre-trained encoder projecting finite-dimensional observations of infinite-dimensional inputs to (finite-dimensional) latent representations, aka intrinsic dimension in the paper, (2) a main network (TEO in the paper), and (3) a pre-trained decoder mapping the latent representation in intrinsic dimension to the data space. The paper claims that the proposed method performs better than previous operator learning methods, including the DeepONet or Neural Operator family.

**Strengths:**

N/A

**Weaknesses:**

In general, I find that the paper’s contribution is not clear.

First of all, it is not clear why the proposed method is a valid operator mapping between infinite dimensional spaces. In my understanding, the key part of the proposed method is “intrinsic dimension calculation”, but this part is somewhat hand-wavy. Moreover, based on the description in the appendix of the algorithm, the intrinsic dimension calculation doesn’t seem like a novel method compared to KNN-based dimensionality reduction algorithms. However, I find that the paper hasn’t provided sufficient information on how the algorithm and the follow-up pipeline can properly map between infinite dimensional spaces.

Second, the presentation of the proposed method is not clear. For instance, the description of each part in the pipeline is hand-wavy, and the paper relies on verbal descriptions instead of what each part does. I find that it is important to describe how each part works in a more concrete manner so that potential audiences can understand how the proposed method can work as an operator.

**Questions:**

N/A

---

> ### Author Response · Authors · 2023-11-15
> **Our response for Reviewer ivj1**
>
> Dear reviewer, we express our sincere gratitude for your review and we have learned your suggestions carefully.
>
>
> **(1) Reply for "It is not clear why the proposed method is a valid operator mapping between infinite dimensional spaces".**
>
> We follow the existed neural operator theory, which derive from Green function kernel integration [1,2]. This theory transforms the operator learning problem into the parameterization problem of kernel integration. Our method focuses on how to choose the appropriate dimension to represent the kernel integral problem by a theory-based method. Thanks for the suggestion, we have incorporated the relevant mathematical proofs into the revised version of our paper.
>
> Through the Navier-Stokes equation with high Reynolds number, we prove that our model can learn operators accurately. In this scenario, our model can generalize different initial condition functions, which means our model can generalize in the Banach space.
>
> **(2) Reply for "the novelty of our intrinsic dimension algorithm".**
>
> The differences between intrinsic dimension calculation and the traditional KNN based dimensionality reduction algorithm can be illustrated by comparing the following formulas:
>
> 1. **Our model (NMO)**.
>    - The latent variables \( $V = \{V_1, \ldots, V_n\}$ \) are regarded as n independently and identically distributed vectors in the latent space \( $R^{d_L}$ \).
>    - The potential variables are bounded on an m-dimensional Riemannian manifold \($M$ \) in \($R^{d_L}$ \), where \($m$ \) is smaller than \($d_L$\).
>    - Use probabilistic methods to estimate the density of distributions of latent variables on the subfluidic form and compute the local intrinsic dimension by maximum likelihood estimation.
>    - Example formula: \( $\log g(f(v))\mu(dv)$ \) Integrates over a ball \( $B(v_0, r)$ \) centered at \( $v_0$ \) with radius \( $r$ \).
>
> 2. **Traditional KNN algorithm**.
>    - The function $KNN(Vt, n neighbors, n jobs) is used to compute the nearest neighbor $.
>    - Convert \( $V_t$ \) to a Numpy array and move to CPU.
>    - Initialize a nearest neighbor object with the given neighbors and jobs.
>    - Return the distance and index of the nearest neighbor.
>
> We take into account the complex structure of the data on potential Riemannian manifolds when dealing with intrinsic dimensions, which is significantly different from traditional KNN methods based on distance computation. While traditional KNN algorithms focus on calculating distances between data points and finding nearest neighbors, our approach focuses more on understanding the intrinsic structure of data through probability distributions and Riemannian geometry. This approach provides a more effective means of dealing with high-dimensional and complex data, especially when considering that the data may exist in low-dimensional manifolds.
>
> **(3) Reply for "The Method part of our paper is verbal."**
>
> Thanks for your review. We've added more mathematical description to method includes proof of learning operator, equivalence between the operator represented by the submanifold and the original operator and optimal dimension.
>
> In sum, we would like to take this opportunity to appreciate your timely help in improving the readability of our paper. We have carefully learned your comments and will improve our paper accordingly. Hopefully, our revised version can address your concerns and make our idea accessible to all readers. Apart from the above responses, you can also read our response to other reviewers for more information.
>
>
> [1] Neural operator: Learning maps between function spaces. *Journal of Machine Learning Research*, 2022.
>
> [2] Fourier neural operator for parametric partial differential equations. *ICLR 2021*.

---

### Official Review · Reviewer_MXPe · 2023-11-05

**Soundness:** 3 good
**Presentation:** 3 good
**Contribution:** 2 fair
**Rating:** 1
**Confidence:** 5

**Summary:**

The authors proposed an encoder/decoder based manifold learning procedure to learn the evolution of a physical system. An extensive experiments have been presented on several datasets with comparative analysis.

**Strengths:**

1. Well written and easy to follow in most parts.
2. The experimental results are illustrative.

**Weaknesses:**

1. My main concern of this paper is the theoretical contribution, in my understanding this paper is essentially learning mapping from data manifold to latent space and used diffusion process on the Euclidean latent space by using transformer, convolution etc.. So essentially the paper has two components (methodologically): (a) deriving the mapping from manifold to Euclidean space. (b) the evolution process. (a) can be learned in a auto-encoder setup, in fact my biggest concern is why authors did not learn the lower-dimensional latent representation instead first learning the higher dimensional manifold and then learning the submanifold?The evolution process is a time-varying recurrent process, hence  very well-explored in past literature.
2. The experimentation in terms of breadth is well appreciated, although I see none of the manifold learning methods used as a baseline. So the comparative analysis is poor.

**Questions:**

1. Why the authors separate encoder and the sub manifold mapping, L. If the authors consider encoder to be a mapping from manifold to lower dimensional sub manifold (i.e., consuming L) is the learning becomes harder?
2. Normally we use latent space to be something lower dimensional, whereas here authors use latent dimension to denote the higher dimensional space where manifold is embedded. This is quite counter intuitive as in practice  researchers use Encoder/Decoder to map from data manifold to latent space (with dimension as intrinsic dimension of the data), as contrary to the higher dimensional space as the authors have claimed to define.
3. Not sure “Therefore, the goal of the manifold algorithm is to calculate minimal m”? For a larger m, we will still assume it is a linear space, it doesn’t have to be minimal to be a linear space. Any higher dimensional space is desired as well. My understanding is the authors try to find a Euclidean latent space where they can use Euclidean stochastic process, so not sure the necessity of finding optimal m.

---

> ### Author Response · Authors · 2023-11-15
> **Our response for Reviewer MXPe**
>
> Dear reviewer, we very appreciate for your careful reading and kind help in reviewing our work. Knowing your efforts and time devoted to comprehensive checking our paper, we have learned your suggestions carefully.
>
> **(1) Why do we separate the autoencoder and the sub-manifold mapping?**
>
> Our paradigm mainly focus on computing the optimal dimensional representation of the underlying operator in the latent space, which requires that we need to train the encoder-decoder structure to learn the mapping function $\mathcal{P}$ and its inverse function $\mathcal{Q}$. This process is trainned by the self-supervised reconstruction strategy. Having trainned the encoder-decoder structure, we can compute the optimal dimension representations for various dynamics by the tensor in the latent space, which process is solving for the intrinsic dimension of the dynamics.
>
> In order to **avoid retraining** the encoder-decoder, we introduce a **non-trainable** linear dimension projection $\mathcal{L}$ to map the original high-dimensional space to the intrinsic dimension. Therefore, we can train the time evolution opeartor module with intrinsic dimension $m$ separately, instead of retraining an encoder-decoder for mapping dimensions to the intrinsic dimension. Otherwise, retraining such an encoder-decoder would result in a wasteful consumption of training resources.
>
> **(2) Why do we set the high dimension mapping for our encoder?**
>
> The core of our intrinsic dimension (ID) algorithm is that computing the minimal dimensional representation for the underlying operator, aiming to prevent dimension redundancy. The process is referred to as describing the degrees of freedom of a physical system in physical dynamics. Employing the minimal degrees of freedom to describe the physical system contributes to the model to better learn the underlying operator[1]. However, before calculating the intrinsic dimensional representation of various physical systems in the latent space, **we cannot ascertain the specific value of these minimal degrees of freedom**. Therefore, we need to map the physical space to a dimension **higher than** the intrinsic dimension to render our paradigm feasible.
>
> **(3) Why we need to find $m$?**
>
> We need to determine the value of $m$ as the dimension for our time evolution operator module to learn the underlying operator representation. If we do not calculate $m$, we will lack a theory for setting the dimension of the evolution operator module. We will further revise the mathematical description of our paper to be more clearly. In our ablation study, we have demonstrated that $m$ is the optimal dimension experimentally.
>
>
> Finally, we would like to thank you for your review.
>
> [1] Automated discovery of fundamental variables hidden in experimental data. *Nature Computational Science*, *2*(7), 433-442, 2022.

---

> > ### Author Response · Authors · 2023-11-15
> > **Our response for Reviewer MXPe (Abaltion Study Table)**
> >
> > Three types implementation of time evolution operator implementations including Multi-Layer Perceptron (MLP), Convolutional Neural Network (CNN) and Transformer (FORMER) structure are evaluated in the ablation experiments. The budget column shows the optimal result for model parameters (PARAM), floating point operations (FLOPs), and training time (TIME) in each dimension. The underline indicates the most accurate result in each implementation. The bold font indicates the intrinsic dimension $m$ of each scenario.
> >
> > | Dimension | MLP (KUROSHIO) | CNN (KUROSHIO) | FORMER (KUROSHIO) | Param (Budget) | FLOPs (Budget) | Time (Budget) | MLP (Navier Stokes) | CNN (Navier Stokes) | FORMER (Navier Stokes) | Param (Budget) | FLOPs (Budget) | Time (Budget) |
> > | --------- | -------------- | -------------- | ----------------- | -------------- | -------------- | ------------- | ------------------- | ------------------- | ---------------------- | -------------- | -------------- | ------------- |
> > | 2         | 0.0430         | 0.0466         | 0.0443            | 1.5414         | 4.9180         | 49.3971       | 0.2557              | 0.2631              | 0.2631                 | 5.2712         | 2.4427         | 8.0935        |
> > | 4         | 0.0429         | 0.0469         | 0.0461            | 1.6922         | 5.0717         | 55.4782       | **0.2547**          | **0.2487**          | **0.2498**             | 5.5496         | 2.5147         | 8.4504        |
> > | 6         | **0.0421**     | **0.0404**     | **0.0427**        | 1.8568         | 5.2375         | 64.8712       | 0.2605              | 0.2593              | 0.2528                 | 5.8418         | 2.5898         | 9.6932        |
> > | 8         | 0.0436         | 0.0471         | 0.0482            | 2.0353         | 5.4154         | 68.3761       | 0.2607              | 0.2637              | 0.2566                 | 6.1478         | 2.6678         | 9.8732        |
> > | 16        | 0.0431         | 0.0484         | 0.0495            | 2.8870         | 6.2474         | 78.6860       | 0.2638              | 0.2677              | 0.2607                 | 7.5098         | 3.0102         | 9.9510        |
> > | 32        | 0.0442         | 0.0511         | 0.0524            | 3.9423         | 7.4721         | 98.3267       | 0.3021              | 0.2655              | 0.2799                 | 8.3212         | 3.6217         | 10.2132       |
> > | 64        | 0.0477         | 0.0490         | 0.0497            | 5.7864         | 8.4464         | 111.8921      | 0.2972              | 0.2802              | 0.3021                 | 9.1213         | 4.2173         | 13.3530       |

---

> ### Author Response · Authors · 2023-11-15
> **Our response for Reviewer MXPe**
>
> **(4) Reply for "Although I see none of the manifold learning methods used as a baseline. So the comparative analysis is poor."**
>
> Manifold learning methods are designed to reduce the dimensionality of data and cannot be directly applied to learn the physical system. Therefore, this is the reason why we do not choose it as the baseline model. Our baseline model has been carefully selected. For instance, our baseline model includes some representative models in time series prediction and state-of-the-art  models in the neural operator, such as the Fourier Neural Operator, as well as the model with strong physical conductive bias named TF-Net. We provide a detailed illustration of baseline model selection in the Experiments section.